# Entwined African and Asian genetic roots of medieval peoples of the Swahili coast

Esther S. Brielle[1✉], Jeffrey Fleisher[2,30✉], Stephanie Wynne-Jones[3,4,30✉], Kendra Sirak[1,5], Nasreen Broomandkhoshbacht[1,5,6], Kim Callan[1,5,6], Elizabeth Curtis[1,5,6], Lora Iliev[1,5,6], Ann Marie Lawson[1,5,6], Jonas Oppenheimer[1,5], Lijun Qiu[5,6], Kristin Stewardson[1,5,6], J. Noah Workman[1,5], Fatma Zalzala[1,5,6], George Ayodo[7], Agness O. Gidna[8], Angela Kabiru[9,10], Amandus Kwekason[8], Audax Z. P. Mabulla[11], Fredrick K. Manthi[12], Emmanuel Ndiema[12], Christine Ogola[12], Elizabeth Sawchuk[13,14,15], Lihadh Al-Gazali[16], Bassam R. Ali[16], Salma Ben-Salem[16], Thierry Letellier[17], Denis Pierron[17], Chantal Radimilahy[18], Jean-Aimé Rakotoarisoa[18], Ryan L. Raaum[19,20], Brendan J. Culleton[21], Swapan Mallick[5,6,22], Nadin Rohland[5], Nick Patterson[1,22], Mohammed Ali Mwenje[23], Khalfan Bini Ahmed[24], Mohamed Mchulla Mohamed[25], Sloan R. Williams[26], Janet Monge[27], Sibel Kusimba[28], Mary E. Prendergast[2,5], David Reich[1,5,6,22,31✉] & Chapurukha M. Kusimba[9,28,29,31✉]

The urban peoples of the Swahili coast traded across eastern Africa and the Indian Ocean and were among the first practitioners of Islam among sub-Saharan people[1,2]. The extent to which these early interactions between Africans and non-Africans were accompanied by genetic exchange remains unknown. Here we report ancient DNA data for 80 individuals from 6 medieval and early modern (AD 1250–1800) coastal towns and an inland town after AD 1650. More than half of the DNA of many of the individuals from coastal towns originates from primarily female ancestors from Africa, with a large proportion—and occasionally more than half—of the DNA coming from Asian ancestors. The Asian ancestry includes components associated with Persia and India, with 80–90% of the Asian DNA originating from Persian men. Peoples of African and Asian origins began to mix by about AD 1000, coinciding with the large-scale adoption of Islam. Before about AD 1500, the Southwest Asian ancestry was mainly Persian-related, consistent with the narrative of the Kilwa Chronicle, the oldest history told by people of the Swahili coast[3]. After this time, the sources of DNA became increasingly Arabian, consistent with evidence of growing interactions with southern Arabia[4]. Subsequent interactions with Asian and African people further changed the ancestry of present-day people of the Swahili coast in relation to the medieval individuals whose DNA we sequenced.

The medieval and early modern Swahili culture of eastern Africa from the seventh century AD was defined by a set of shared features: a common language of African origin (Kiswahili), a shared predominant religion (Islam) and a geographic distribution in coastal towns and villages. People of the Swahili culture lived across a vast coastal region that included northern Mozambique, southern Somalia, Madagascar and the archipelagos of Comoros, Kilwa, Mafia, Zanzibar and Lamu[1] (yellow outlines in Fig. 1a). Millions of present-day coastal people identify as Swahili, although for many this is a secondary identity, with primary identities often being more based on town of origin, family history or traditional social status[5]. How people who identify as Swahili in the present day relate to people of the medieval and early

[1]Department of Human Evolutionary Biology, Harvard University, Cambridge, MA, USA. [2]Department of Anthropology, Rice University, Houston, TX, USA. [3]Department of Archaeology, University of York, York, UK. [4]University of South Africa, Pretoria, South Africa. [5]Department of Genetics, Harvard Medical School, Boston, MA, USA. [6]Howard Hughes Medical Institute, Harvard Medical School, Boston, MA, USA. [7]Jaramogi Oginga Odinga University of Science and Technology, Bondo, Kenya. [8]National Museums of Tanzania, Dar es Salaam, Tanzania. [9]Department of Archaeology, National Museums of Kenya, Nairobi, Kenya. [10]British Institute of Eastern Africa, Nairobi, Kenya. [11]Department of Archaeology and Heritage Studies, University of Dar es Salaam, Dar es Salaam, Tanzania. [12]Department of Earth Sciences, National Museums of Kenya, Nairobi, Kenya. [13]Cleveland Museum of Natural History, Cleveland, OH, USA. [14]Department of Anthropology, University of Alberta, Edmonton, Alberta, Canada. [15]Department of Anthropology, Stony Brook University, Stony Brook, NY, USA. [16]Department of Genetics and Genomics, College of Medicine and Health Sciences, United Arab Emirates University, Al-Ain, United Arab Emirates. [17]Laboratoire Evolution et Santé Orale, Faculté de Chirurgie Dentaire, Université Toulouse III—Paul Sabatier, Toulouse, France. [18]Institut de Civilisations/Musée d'Art et d'Archéologie, Université d'Antananarivo, Antananarivo, Madagascar. [19]Department of Anthropology, Lehman College and The Graduate Center, The City University of New York, New York, NY, USA. [20]The New York Consortium in Evolutionary Primatology, New York, NY, USA. [21]Institutes of Energy and the Environment, The Pennsylvania State University, University Park, PA, USA. [22]Broad Institute of Harvard and MIT, Cambridge, MA, USA. [23]National Museums of Kenya, Lamu Museums, Lamu, Kenya. [24]Coastal Archaeology, Gede National Monument, Gede, Kenya. [25]Coastal Archaeology, Fort Jesus Museum, Mombasa, Kenya. [26]Department of Anthropology, University of Illinois at Chicago, Chicago, IL, USA. [27]University of Pennsylvania Museum of Archaeology and Anthropology, Philadelphia, PA, USA. [28]Department of Anthropology, University of South Florida, Tampa, FL, USA. [29]Institute of African Studies, University of Nairobi, Museum Hill, Nairobi, Kenya. [30]These authors contributed equally: Jeffrey Fleisher and Stephanie Wynne-Jones. [31]These authors jointly supervised this work: David Reich and Chapurukha M. Kusimba. ✉e-mail: ebrielle@fas.harvard.edu; jfleisher@rice.edu; stephanie.wynne-jones@york.ac.uk; reich@genetics.med.harvard.edu; ckusimba@usf.edu

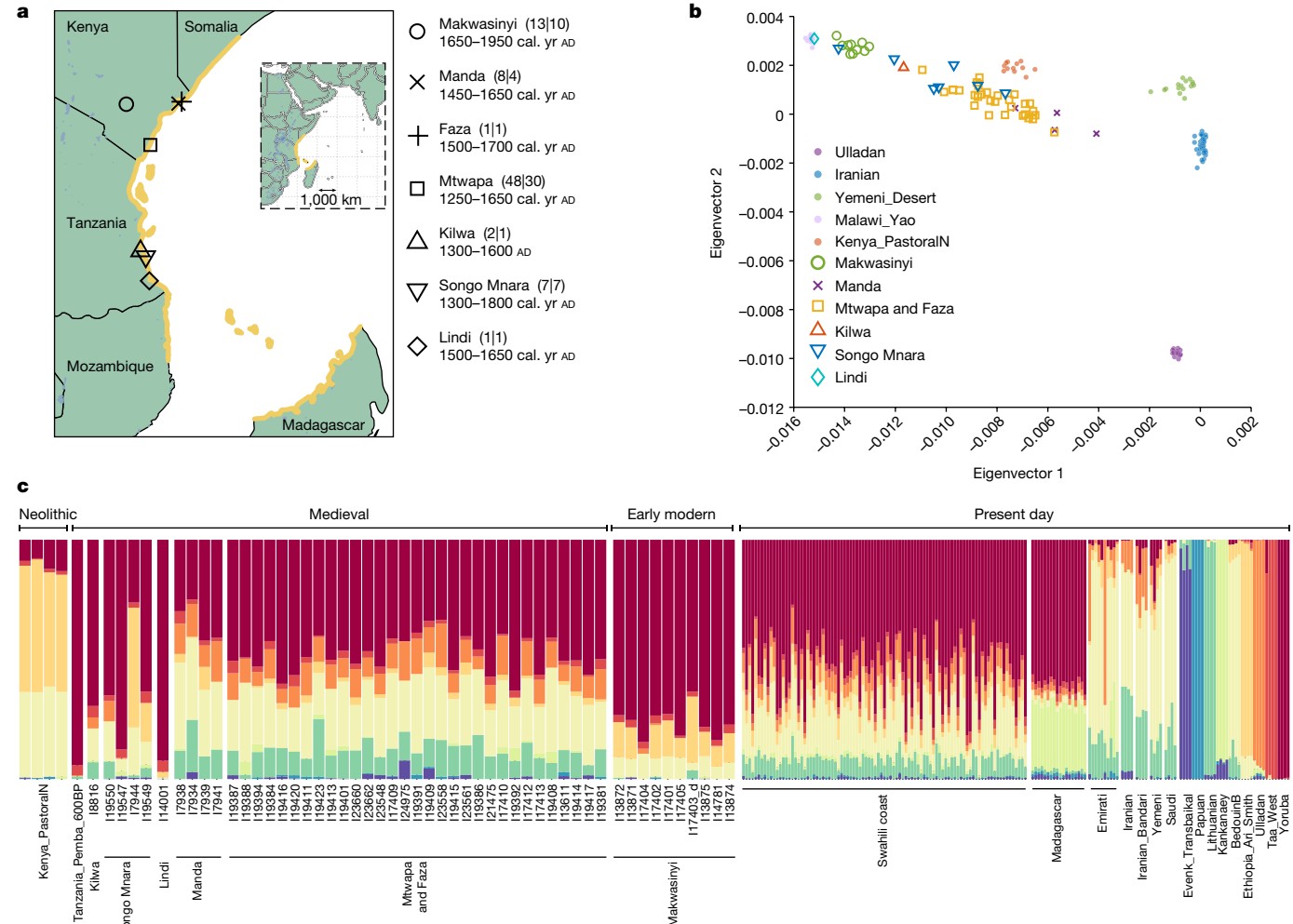

**Fig. 1 | Dataset overview. a**, Coastal areas associated with the medieval Swahili culture are shown in yellow. Sites represented in the ancient DNA samples are marked with black shapes. Numbers in parentheses are formatted *X|Y*, where *X* is the number of individuals for whom there are data, and *Y* is the number of individuals for whom we report high-resolution analyses. The chronology is given as the union of 95% confidence intervals for direct radiocarbon dates on the skeletons rounding to the nearest 50 years and is shown as calibrated (cal.) yr AD, or as AD for sites with only archaeological context (Supplementary Information; Extended Data Table 1). The base map was made with the Natural Earth R package using a CC BY license. Bodies of water were added from the RCMRD Geoportal with a CC BY license in R. All points were added and modifications were performed in R and Adobe Illustrator. The map is published for the first time in this Article. **b**, In a principal component analysis, eigenvector 1 correlates to variation maximized in sub-Saharan Africa, and eigenvector 2 correlates to Eurasian variation. **c**, Ancestry component assignment using ADMIXTURE with *K* = 9 clusters (selected on the basis of low cross-validation errors, high log-likelihood scores, and a low number of reference populations to not overfit; groups that maximize each components are shown on the right). Individuals with sufficient data for high-resolution analysis are plotted in approximate chronological order from left to right. Ancient individuals are labelled and plotted at four times the width of present-day individuals.

modern Swahili culture has been difficult to elucidate in the absence of ancient DNA.

The medieval largely autonomous towns and polities known as the Swahili states arose out of fishing and agropastoral settlements on the eastern African coast during the late first millennium AD[6]. First millennium AD sites on the littoral, beginning in the seventh century, were part of a shared material culture and practice network across the eastern African region[7]. These sites were engaged in the Indian Ocean trading system, facilitated by southwest monsoons from May to October that enabled merchant vessels to travel from India or the Arabian Peninsula to the eastern African coast, and northeast monsoons from November to March enabling their return in the same year[8].

Muslims were present from the eighth century AD, probably as a minority[9]. A major archaeological transition is evident during the eleventh century, with the establishment of new settlements and the elaboration of older ones with coral-built mosques and tombs, a set of changes generally understood as coinciding with the widespread adoption of Islam[10]. At this time, clearer distinctions also emerged between coastal ceramics and material traditions and those of inland assemblages[2,11], even as many aspects of material culture remained deeply linked with inland African groups.

The political and administrative independence of Swahili towns diminished in the sixteenth century as the Portuguese naval and economic dominance in the Indian Ocean spread[12]. In the early eighteenth century, the Portuguese influence waned and the Sultanates of Oman and later Zanzibar became dominant[4]. In the nineteenth century, the growth of overseas trade, including in enslaved people, led to large-scale population movements from central regions of Africa and settlers from the Yemeni region of Hadramawt[4,13]. In the mid-nineteenth century, Britain and other European powers became dominant, leading to the settlement of Europeans, the arrival of labourers from South Asia, and further interactions with non-coastal eastern Africans.

In light of this multi-layered history, the extent to which people who identify as Swahili in the present day are genetically linked with people who built the medieval trading towns is unclear, as is the relationship of the medieval groups to earlier groups. Although the intercontinental connections maintained as part of the Indian Ocean trading network meant that foreigners were consistently present along the coast, the extent to which they had families with African residents has long been debated[14].

Swahili traditions suggest that foreigners had an important impact; a set of common oral histories relates the founding of coastal towns to the arrival of a group known as the Shirazi, referring to a region in Persia[3]. This Shirazi tradition was put into writing in the Kilwa Chronicle in the sixteenth century[15]. These accounts of Shirazi roots were central to the narrative constructed by mid-twentieth century colonialist archaeologists, who interpreted second millennium coastal eastern African sites as having been built by Persian and Arab settlers, and focused on connections with the broader Indian Ocean world[16].

However, narratives of foreign origin have the potential to be misleading, as Swahili social 'elites' used claims of foreign origin and rejection of cultural connections within Africa to establish their social status and to signal their religious and cultural affinities[17,18].

Recent research has shown that archaeology during colonial times tended to ignore the evidence of deep African roots, emphasizing foreign objects at medieval Swahili sites rather than providing a balanced picture of the archaeological record[2]. Imports at most coastal sites typically comprise less than 5% of total assemblages[2,9]. Other aspects of the material culture also show continuity with earlier settlements, including the persistence of crops, domesticated animals, craft styles and ceramics[9,19]. Linguistic evidence provides additional evidence of African roots: Kiswahili is an African Bantu language with Asian loanwords[20]. However, without ancient DNA evidence, it is not possible to directly address questions of how genetic ancestry changed over time.

We generated ancient DNA data from the skeletal remains of individuals found at six coastal or island towns: Mtwapa, Manda, Faza, Kilwa, Songo Mnara and Lindi. These individuals date to AD 1250–1800 but provide insight into genetic events from the tenth century AD onwards. We also generated ancient DNA from the remains of individuals found at the site of Makwasinyi (postdating around AD 1650), about 100 km inland from the southern Kenyan coast, which was inhabited by people who were in cultural contact with coastal groups. We compare the newly reported data from the ancient individuals with that of present-day coastal Swahili speakers and with previously published data from diverse ancient and present-day eastern African and Eurasian groups.

## Dataset overview

We generated 179 ancient DNA libraries from 156 distinct skeletal samples (Supplementary Data File 1 and Methods). We applied in-solution enrichment for a targeted set of about 1.2 million single nucleotide polymorphisms (SNPs) to obtain genome-wide data passing standard measures of ancient DNA authenticity from 80 distinct individuals. The individuals were buried at seven second millennium AD sites in eastern Africa (black shapes in Fig. 1a; individuals are listed in Extended Data Table 1 and Supplementary Data File 2; see Methods, 'Inclusion and ethics' and Supplementary Information for details of archaeological and genetic permissions and sampling). We obtained direct radiocarbon dates (Supplementary Data File 3) for 33 of the skeletons, and estimated date ranges for the other individuals on the basis of archaeological context or genetic evidence of relatedness to individuals for whom we had direct dates (Extended Data Table 1 and Supplementary Data File 2). Because of the reliance on seafood, old carbon entering the food chain (marine reservoir effects) could mean that the radiocarbon dates of some individuals are older than their true dates. Moreover, differences in the dependence on marine food across the archaeological sites could mean that the relative chronology of the coastal individuals and sites may not be possible to determine with full confidence. We also generated new genome-wide data on the Affymetrix Human Origins SNP array from 93 present-day individuals who identified as Swahili and indicated that their ancestors lived for many generations in coastal towns[21] (Supplementary Data File 4). Finally, we generated new genome-wide data from 19 individuals from Madagascar, and 10 from the United Arab Emirates.

Three sites were northern coastal towns: Mtwapa (Supplementary Fig. 1, 48 individuals spanning AD 1250 to AD 1650), Faza on Pate Island (1 individual), and Manda Island (Supplementary Fig. 2, 8 individuals spanning AD 1450 to AD 1650). Three additional sites were southern coastal towns: Songo Mnara (Supplementary Fig. 3 and Supplementary Table 1, 7 individuals spanning AD 1300 to AD 1800), Lindi (1 individual at AD 1500 to AD 1650), and Kilwa Kisiwani (2 individuals spanning AD 1300 to AD 1600). The remains at Mtwapa, Manda and Songo Mnara were mainly from Muslim burials of elites, often located near mosques. We do not have enough context for the Faza, Kilwa and Lindi burials to know if they followed the same pattern. We also analysed 13 individuals from Makwasinyi (AD 1650–1950), approximately 100 km inland from the coast of present-day Kenya. Although these burials post-date the coastal sites, the Makwasinyi community traded with coastal peoples while remaining isolated in most respects. We hypothesized that the ancestry of Makwasinyi people might be a good proxy to represent inland African groups that may have been in contact with people from medieval towns on the northern Swahili coast in previous centuries[22].

Of the 80 individuals for whom we report data, we exclude 26 in genome-wide analyses, although their data remain valuable. Of these, 18 had too few SNPs for high-resolution whole-genome analyses although they yielded useful data such as reliably determined mitochondrial sequences; 5 were first- or possibly second-degree relatives of other individuals in the dataset with higher-quality data; 2 showed evidence of contamination; and 1 was a population genetic outlier with limited data, raising the possibility of contamination (Supplementary Data File 2).

Ancient DNA data from four individuals from the eastern African coast have previously been published (Supplementary Data File 5), but none have been published from a Swahili town[23]. An individual from around AD 1400 whose remains were recovered from Makangale Cave on Pemba Island had ancestry predominantly related to western African groups[23] (an ancestry common today in speakers of Bantu languages and prevalent in eastern Africa—hereafter referred to as 'Bantu-associated'). Another individual from Makangale Cave on Pemba Island dated to around AD 600, an individual from around AD 600 from Kuumbi Cave on Zanzibar Island, and an individual from around AD 1500 from Panga ya Saidi in Kenya all had predominantly sub-Saharan African forager-associated ancestry[23]. There is no indication of Eurasian ancestry deriving from migrations in the last 2,000 years in any of these individuals, which differs from nearly all of the individuals from medieval coastal towns newly reported here.

In this Article, we use 'African ancestry' to refer to DNA deriving from people who can be genetically well-proxied by sub-Saharan Africans for whom there is published ancient DNA data dating to between 2000 BC and AD 1000. We use the terms 'Eurasian', 'Persian', 'Arabian' and 'Indian' to refer to ancestry that can be proxied by modern populations from these regions and that are not known to be similar to ancestry in sub-Saharan Africans between 2000 BC and AD 1000. The evidence that a proportion of the ancestors of Africans from between 2000 BC and AD 1000 may have come from Eurasia—for example, approximately 40% of the ancestry of people of the eastern African Pastoral Neolithic culture[24]—does not contradict these definitions, as all humans are mixed at multiple time depths of history. As long as we specify both the time and the geography for the source populations, we can be precise in the use of the term 'African ancestry'[25].

## Genetic affinities

To obtain a qualitative picture of the sources of the ancestry in the ancient individuals, we performed principal component analysis (PCA) (Fig. 1b). We used 1,286 present-day Eurasian and African individuals to compute the axes (Supplementary Data File 6). We projected the newly reported ancient individuals onto this PCA and found that they form a cline, with one end overlapping with ancient and present-day African groups and the other falling between present-day Persians and Indians. This suggests mixtures of different proportions of source populations at either end of the cline, with these sources potentially having multiple deeper ancestry components. Some coastal individuals—particularly from Songo Mnara and Lindi, do not fall on this cline—suggesting additional complexity, although our power to understand this variability is limited by the small sample sizes. Similar patterns are evident with unsupervised clustering using ADMIXTURE, which further suggests sub-Saharan African-associated components, southwest Asian-associated components, and East Asian or Indian-associated components (Fig. 1c and Extended Data Fig. 1).

## Proportions of African, Persian and Indian DNA

Using qpAdm[26], we find that most medieval and early modern individuals can only be fit by a model with at least three ancestry components that can be proxied with ancient African, present-day Iranian and present-day Indian populations (Fig. 2a, Extended Data Table 2, Supplementary Tables 2–10 and Supplementary Information). Such a three-source model fits the pool of 48 Mtwapa individuals and the Faza individual (*P* value for fit = 0.23); the pool of Manda (*P* = 0.28) individuals, and at least one Songo Mnara individual (I19550) (*P* = 0.38). One Kilwa individual (I8816) had a relatively high proportion of ancestry related to inland sub-Saharan Africans, and so the Indian proportion, which is the smallest contributor, falls below the threshold of definitive detection (a two-source model fits; *P* = 0.27).

The type of African ancestry needed to make the models fit differed between individuals from the north (Mtwapa, Faza and Manda) and south (Kilwa and Songo Mnara) of the studied region. In Kenya, the best-fitting proxy African source is the inland Makwasinyi individuals (Extended Data Table 2), who are themselves well-modelled as mixtures of about 80% Bantu-associated and 20% ancient eastern African Pastoral Neolithic ancestry[24] (Fig. 3a, Extended Data Table 3, Supplementary Tables 2 and 3 and Supplementary Information). In Tanzania, the best-fitting African proxy source is Bantu-associated without evidence of a Pastoral Neolithic contribution. We use the individual buried at around AD 1600 in Lindi as a proxy Bantu-associated source for the Kilwa individual and individual I19550 from Songo Mnara (Extended Data Table 2).

Although three continental sources are required to fit the data, the individuals from Manda, Faza and Mtwapa form a cline in PCA, suggesting two proximal source populations (Fig. 2b). Using linear regression, we extrapolate the ancestry of these two sources and infer that one was consistent with a 100% African origin (Supplementary Fig. 5 and Supplementary Information). The same analysis concludes that the other source had both Persian and Indian ancestry. This is consistent with sub-Saharan Africans mixing with a group that already had a mixture of Persian and Indian ancestry components. Given the two different African sources for the northern and southern individuals, there must have been at least two, but possibly more, admixture events. This would be expected if people of mixed Persian and Indian ancestry had children with people from different local African populations at different locations along the coast.

When we analyse the Mtwapa and Faza, Manda, Kilwa and Songo Mnara individuals separately, the estimated proportions of Eurasian ancestry from India overlap, which could be consistent with a homogeneous source population of mixed Indian–Persian ancestry for all sites (Fig. 2c). However, a variable proportion among the early immigrants cannot be ruled out, and thus we cannot distinguish between scenarios of two or more streams of Persian–Indian migrants. Our statistical power to detect Indian ancestry relies on pooling of data from multiple individuals. There are a number of individuals with limited data or low Eurasian ancestry, for example individual I8816 from Kilwa, for which we have little power to detect Indian ancestry and cannot definitely document it (Supplementary Information).

## Males from Persia and females from Africa

We tested whether male and female ancestors contributed the same proportions of African-like, Persian-like and Indian-like ancestry to ancient individuals in the northern coastal sites and Kilwa (Table 1). To carry out this analysis, we used the fact that chromosomes 1–22, the X chromosome, mitochondrial DNA (mtDNA) and the Y chromosome are passed down to subsequent generations in different ways by males and females. We could not perform the same analysis at Songo Mnara because no individuals with high quality data fit the three-way model.

We first analysed mtDNA (Extended Data Table 1 and Table 1). Analysing 62 individuals with confidently determined mtDNA haplogroups (including relatives and individuals with low coverage genome-wide data), we find that 59 carry an L* haplogroup, which in the present day is almost entirely restricted to sub-Saharan Africans[27]. The exceptions are a pair of first- or second-degree relatives from Mtwapa carrying M30d1, which in the present day is largely restricted to South Asia[27], and an individual with haplogroup R0+16189, which today is characteristic of Saudi Arabia and the Horn of Africa[28]. These results are consistent with female ancestry deriving overwhelmingly from African sources.

Analysing male-transmitted Y chromosome DNA, we find that two out of three non-first-degree related males from Manda carry haplogroup J2, and the third carries G2. Both haplogroups are characteristic of Southwest Asia (plausibly Persia) and are largely absent in sub-Saharan Africans[29]. The Kilwa individual also carries J2. Fourteen out of 19 males from Mtwapa have Y chromosome haplogroups in the J family, and two are of the R1a haplogroup, all considered typically non-African. Only 3 out of 19 Mtwapa males, along with the Faza male, are in the E1 family characteristic of sub-Saharan Africa.

We next compared chromosome X, which occurs as two copies in females and one in males and so reflects mostly female history, to the autosomes (chromosomes 1–22), which equally reflect female and male history (Methods and Supplementary Information). Chromosome X estimates of African ancestry are higher than on the autosomes at all sites, providing an independent line of evidence that African ancestry is primarily from females and Persian ancestry is primarily from males (Table 1). Assuming that the mixture occurred over just a few generations, we obtain quantitative estimates of the proportion of African ancestry from females as 100% at Manda, 69–97% at Mtwapa-Faza, and 69–100% at Kilwa (Methods and Table 1). We estimate Persian ancestry at Mtwapa-Faza and Kilwa to be 100%, and at Manda as 90–100%. If the mixture occurred over more generations, we cannot obtain a point estimate, but can nevertheless infer primarily African female and Persian male sources.

Together, these multiple lines of evidence show that the Southwest Asian ancestors of the Mtwapa and Faza, Manda and Kilwa individuals were almost entirely male, whereas the African ancestors were almost entirely female.

## Mixing began by AD 1000

We estimated when mixture occurred on the basis of the sizes of stretches of ancestry inherited from the ancestral populations, which break up at a known rate every generation[30]. We calculated 95% confidence intervals for the inferred dates of AD 795–1085 for a pool of the northern Mtwapa, Faza and Manda individuals, and AD 708–1219 for a pool of the southern Kilwa and Songo Mnara I19550 individuals (Table 1 and Extended Data Fig. 3). The uncertainty intervals overlap

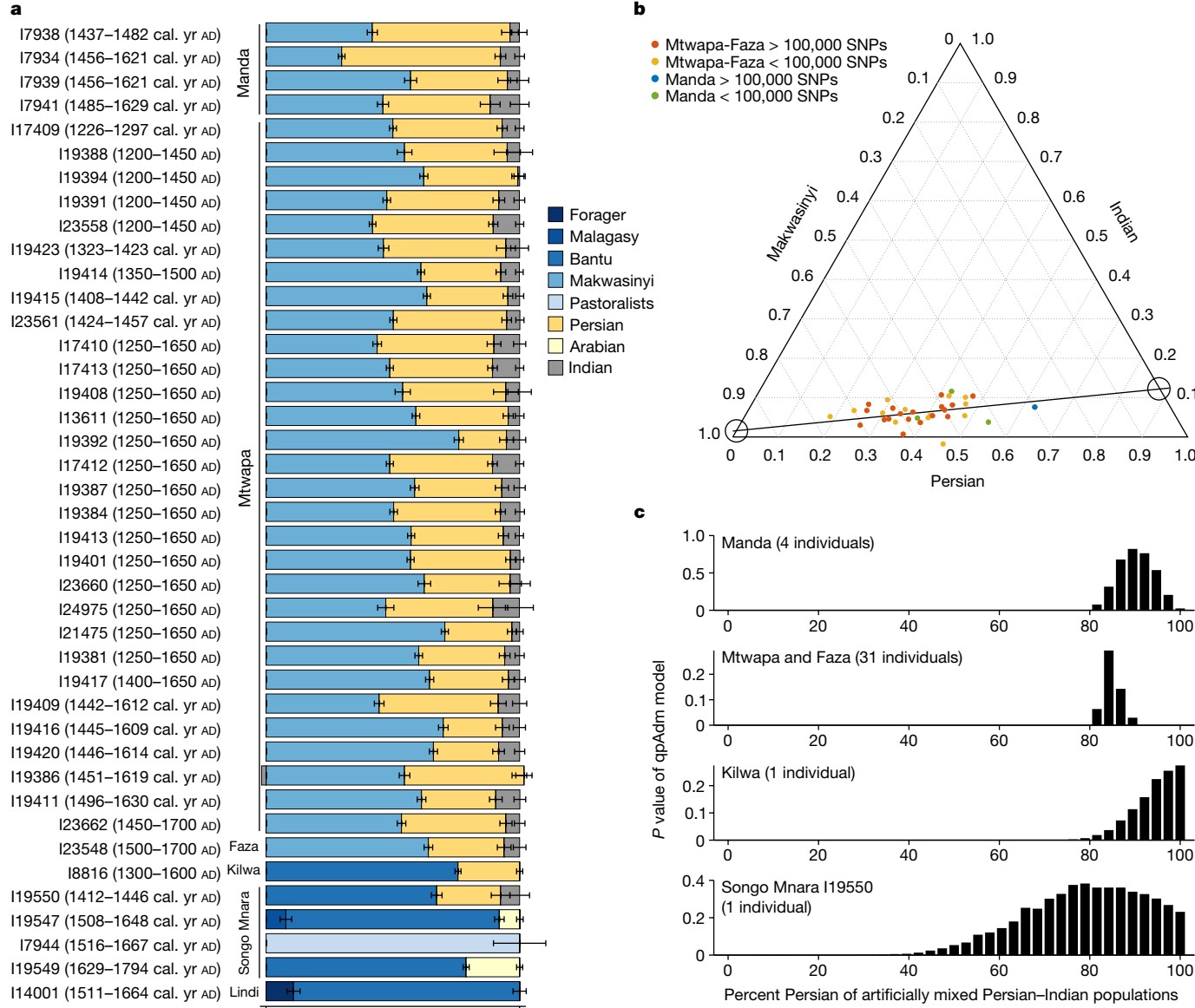

**Fig. 2 | Individual ancestry proportions. a**, Inferences from qpAdm (see Extended Data Table 2 and Supplementary Information for model details and statistical fit). Blue represents African ancestry: the most common are Bantu-associated (common at southern sites) and Makwasinyi associated (northern sites), which itself is approximately 80% Bantu-related and 20% pastoralist-related. Yellow represents Southwest Asian ancestry: Persian or Arabian. Grey represents Indian ancestry. Bars represent s.e.m., computed using a block jackknife across all 5-centimorgan (cM) segments of the autosomes, and are meaningful even for single individuals as the genome contains information from a large sample size of ancestors. **b**, Ternary plot of

Makwasinyi, Persian and Indian ancestry components in Mtwapa and Faza (red (high coverage) and yellow (low coverage)) and Manda (blue (high coverage) and green (low coverage)). Individuals with higher coverage (>100,000 SNPs overlapping positions on the Human Origins SNP array) are used to fit a linear regression (dashed line), which intersects at nearly 100% Makwasinyi and 0% Persian and Indian, consistent with a Makwasinyi-related population with little or no recent Asian ancestry mixing with an already-mixed Persian–Indian population. **c**, Bar graph showing *P* values from Hotelling *T*-squared tests for a qpAdm model with a mixed Persian–Indian source. The *x*-axis specifies the proportion of Persian ancestry in the source.

from AD 795 to AD 1085. These estimates would be biased too old if there was a marine reservoir effect. The inferred dates also reflect an assumption that the mixture occurred all at once; however, mixture of Eurasians and Africans was certainly drawn out over a number of generations, and indeed the historical evidence and our genetic analysis that follows document continued incorporation of migrants from both inland Africa and Eurasia until the present. However, simulations show that mixture must have begun by the inferred date[31], and thus we can be confident that already-mixed males with both Indian and Persian ancestry were present along the coast by around AD 1000, and began mixing with primarily female sub-Saharan Africans by that time.

## Arabians and other migratory influences

Although almost all the coastal individuals we analysed had Asian ancestry, there were exceptions. Some early modern individuals at Lindi and Songo Mnara showed no evidence of recent Asian ancestry (I14001 and I7944) (Extended Data Fig. 2a, Extended Data Table 3, Supplementary Table 11 and Supplementary Information). We find possible Malagasy-associated ancestry in Songo Mnara (I19547) (Extended Data Fig. 2a, Extended Data Table 3 and Supplementary Information). Our finding of coastal individuals who differ from others from similar times or regions attest to continued exchange with people in the Indian Ocean trading network, although our sample size is too small to identify general patterns.

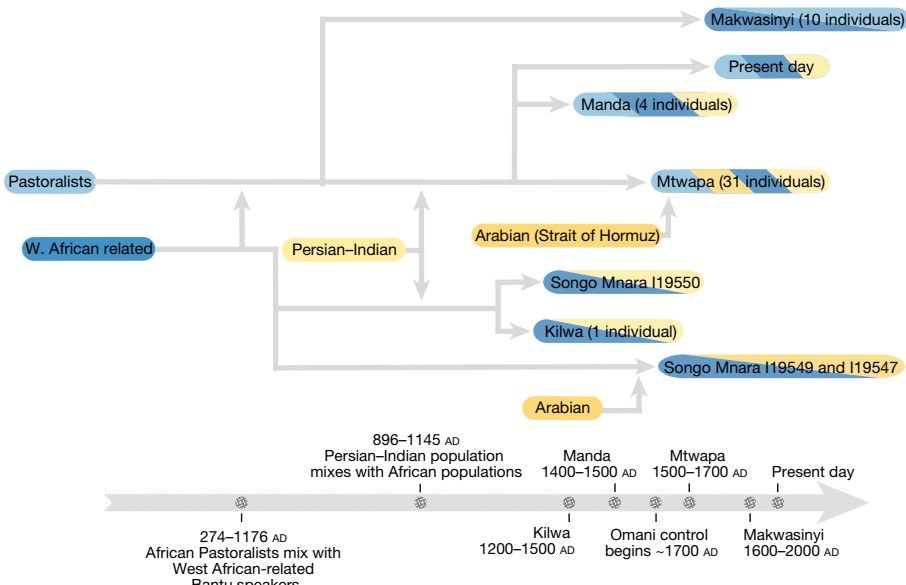

**Fig. 3 | Inferred admixture events along the eastern African coast.** African populations are represented in shades of blue; Southwest Asian and Indian populations are represented in shades of yellow. Populations with both colours represent those that are admixed between the corresponding proxy source populations.

For some of the individuals in our study with Asian ancestry (from Manda, Kilwa and I19550 from Songo Mnara), there is only evidence of Persian or Persian–Indian ancestry. However, other individuals, particularly from Mtwapa, can only be modelled by sources with some Arabian-associated ancestry when using other Mtwapa or Manda individuals as a source (Supplementary Fig. 6, Supplementary Tables 12–15 and Supplementary Information). We were unable to determine the exact source of the Arabian-associated ancestry. However, we know that it is somewhere on the genetic gradient between Arabians and Persians. A proxy source that provides a fit for this Arabian-related ancestry at Mtwapa is present-day people who live on the shores of the Strait of Hormuz, which separates the Arabian Peninsula from Iran. The Strait of Hormuz and the Swahili coast were under Omani control by the end of the seventeenth century.

Direct genetic evidence for Arabian-associated migration comes from two individuals from Songo Mnara. Both date to the early modern period when contacts with Arabia are well documented, and they can only be modelled with Arabian-related ancestry in qpAdm (Fig. 2a). Analyses of present-day coastal populations also point to Arabian genetic influences. Whereas the Asian ancestry of some individuals can be modelled entirely as Persian–Indian as in medieval Manda, for other individuals, ancestry from the Strait of Hormuz is a better fit (coloured salmon in Supplementary Data File 7).

## Relating medieval to modern Swahili

We assembled genome-wide data for two groups of present-day people who identify as Swahili: 89 with previously reported data[32] and 93 for which we generated new data.

Most of the individuals in the previously published dataset (87 out of 89) have only a modest inheritance from people with ancestry resembling the medieval people that we sampled from nearby coastal areas (Extended Data Fig. 2a, Extended Data Table 3, Supplementary Table 16 and Supplementary Information). We estimate that 11 ± 4% have medieval Swahili-associated ancestry; 84 ± 3% have Bantu-associated ancestry; and 6 ± 3% have Pastoral Iron Age-associated ancestry[24] (Extended Data Fig. 2a and Extended Data Table 3). These patterns are mirrored in the Y chromosome haplogroups, with 95% having typically African haplogroups—in contrast to the medieval coastal individuals, for whom almost all Y haplogroups are associated with Near Eastern

## Table 1 | Population mixture and evidence of sex bias in three groups from the Swahili coast

| | Mtwapa | | | Manda | | | Kilwa | |
| --- | --- | --- | --- | --- | --- | --- | --- | --- |
| | **Africa** | **Persia** | **India** | **Africa** | **Persia** | **India** | **Africa** | **Persia** |
| mtDNA haplogroups (N) | 29 | - | 1 | 3 | - | - | 1 | - |
| Y chromosome haplogroups (N) | 4 | 16 | - | - | 3 | - | - | 1 |
| Ancestry sources, autosomes | 56–58% | 34–39% | 4–9% | 30–34% | 55–65% | 3–13% | 72–76% | 24–28% |
| Ancestry sources, X chromosome | 65–75% | 0–24% | 7–39% | 49–70% | 0–74% | 0–75% | 83–100% | 0–17% |
| Z-score autosomes vs X chromosome | −4.7 | 3.4 | −2.0 | −5.0 | 1.4 | −0.5 | −3.3 | 3.3 |
| Female ancestry | 69–97% | 0% | 33–100% | 100% | 0–10% | 0–100% | 69–100% | 0% |
| Male ancestry | 3–31% | 100% | 0–67% | 0% | 90–100% | 0–100% | 0–31% | 100% |
| Persian (proportion of total Asian) | 84–87% | | | 84–97% | | | 89–100% | |
| Estimated admixture dates | AD 795–1085 | | | | | | AD 708–1219 | |

We compared ancestry estimates on chromosomes 1–22 and the X chromosome to infer proportions from female or male ancestors (Methods). Ranges are 95% confidence intervals, truncated if extending outside 0–100%. The proportion of ancestry from one sex was set to 0% or 100% if the range fell beyond these limits.

## Table 2 | Comparison of haplogroup distributions in medieval Swahili individuals with those from two studies of present-day Swahili people

| Likely Y chromosome haplogroup origin | Present-day sampled in ref. [21] and published here | Present-day published previously[32] | Medieval |
|---|---|---|---|
| Africa | 45% | 95% | 17% |
| Southwest Asian | 36% | 3% | 83% |
| Other | 20% | 2% | 0% |
| **Likely mtDNA haplogroup origin** | | | |
| Africa | 94% | 97% | 97% |
| South or East Asia | 4% | 2% | 3% |
| Other | 2% | 1% | 0% |

people (Tables 1 and 2). However, the newly generated data show a much higher inferred proportion of ancestry from groups similar to the medieval ones, ranging from 46 to 77% medieval Swahili-associated ancestry (depending on whether we use a pool of Manda and Mtwapa individuals as the sources) (Extended Data Table 4 and Supplementary Data File 7). Y chromosome haplogroups in the new data are also consistent with a greater contribution from medieval peoples: the African-associated Y-haplogroup frequency was 45%, much larger than the 17% in the medieval individuals, but smaller than the 95% estimated in the previously published study (Table 2).

The differences in the proportions of ancestry in the present-day individuals identifying as Swahili may reflect differences in how they were estimated. The previously published dataset included people from the coastal towns of Kilifi, Lamu and Mombasa in Kenya who indicated that their family had been Swahili-speaking for the past three generations. The newly published dataset included people from 13 locations along the Kenyan coast who indicated that their ancestors had lived in coastal towns and had a Swahili identity for many generations, thus enriching for more traditional upper-class Swahili people who plausibly retained more ancestry from medieval coastal individuals[30]. The greater medieval coastal ancestry may also reflect isolation. In the newly published dataset from the mainland, individuals from the site of Jomvu Kuu had significantly less medieval coastal ancestry than the individuals from the other sites, all of which were from islands that were plausibly more isolated from admixture with inland groups ($P = 3.3 \times 10^{-6}$, one-sided Wilcoxon rank sum test).

We used the lengths of African and Persian genetic segments to estimate the age of admixture in the newly published modern individuals[30] at AD 1096–1410. These data are more recent than the medieval coastal samples, consistent with ongoing mixture with African or Asian populations since medieval times.

## No recent Asian ancestry in inland people

The Makwasinyi individuals date to the past three centuries, and are from deep in the Tsavo region, nearly 50 km from the nearest population centre. The Makwasinyi group fits as a proxy source for African ancestry in qpAdm modelling of the Mtwapa, Faza and Manda individuals, but unlike these individuals, qpAdm finds no evidence of recent Asian ancestry in the Makwasinyi group (similar to present-day non-coastal populations; Supplementary Table 17). Instead, Makwasinyi individuals are similar in ancestry to the modern individuals identified as Swahili in the previously reported dataset[32]. They are well-modelled as $21.3 \pm 1.2\%$ Pastoral Neolithic-associated ancestry (from herders present in eastern African after 3000 BC) and $78.7 \pm 1.2\%$ Bantu-associated ancestry (from farmers present after 1000 BC) (Extended Data Fig. 2a). We did not detect sex bias in the history of the formation of this population (Methods, Supplementary Information). The average date for the

Bantu-Pastoral Neolithic-associated mixture is around AD 300–1200 (Extended Data Fig. 3), with most of this range consistent with the archaeological evidence for the impact of the Bantu expansion on this region.

## Discussion

A key finding of this study is evidence of mixture at roughly AD 1000 between peoples of African and Persian ancestries (Fig. 3 and Table 1). This is consistent with the Kilwa Chronicle, which describes the arrival of Persians on the Swahili coast and interactions between them and coastal residents. Whether or not this history has a basis in an actual voyage, the ancient DNA provides direct evidence for Persian-associated ancestry being derived overwhelmingly from males and arriving on the eastern African coast by about AD 1000. This timing coincides with archaeological evidence for a substantial cultural transformation on the coast, including the widespread adoption of Islam[10]. At Kilwa, coin evidence has dated a ruler linked to a Shirazi (Persian) dynasty, Ali bin al-Hasan, to the mid-eleventh century[33]. The genetic evidence suggests that this arrival was accompanied by mixture, which began by AD 1000, and continued later. People of both African and Asian ancestry made major contributions, with African proportions of approximately 57% on average at Mtwapa and Faza, 32% at Manda, 67% at Songo Mnara and 74% at Kilwa (Table 1, Extended Data Table 2 and Supplementary Table 8).

Archaeological evidence provides important context for our genetic findings. The individuals that we analysed lived in the thirteenth to eighteenth centuries, and were excavated mostly from elite contexts. However, coastal sites from around AD 1000, when the mixture occurred, showed little evidence for distinct societal elites. Three of the sites sampled here (Mtwapa, Songo Mnara and Faza) did not exist as towns in AD 1000, and so these admixed populations moved to those towns later. Thus, the elite inhabitants of Mtwapa and other sites developed from admixed populations and were not foreign migrants or colonists.

Linguistic evidence provides further context. Kiswahili is a Bantu language, and since most ancestry in medieval Swahili people derives from African people, our results suggest that the children of immigrant men of Asian origin adopted the languages of their mothers, a common pattern in matrilocal cultures[34]. However, Kiswahili also has non-African influences, reflecting a millennium and a half of intense interaction with societies around the Indian Ocean rim. Persian loanwords contribute up to 3% of Kiswahili, but it is unclear whether they are derived directly from Persian or through adoption into other Indian Ocean languages[35]. Arabic loanwords are the single largest non-Bantu element in Kiswahili[35] (16–20% of words), and may be primarily due[20] to incorporations after AD 1500.

A recurrent theme of our findings is the different participation of males and females in population mixture events. We find evidence of predominantly male Southwest Asian ancestors mixing with predominantly female African and, to a much lesser extent, female Indian ancestors in the lineages of medieval people on the Swahili coast. This provides evidence for asymmetric social interactions between groups as cultural contact occurred, although such genetic data cannot reveal the processes contributing to these patterns.

This study provides information from only a subset of times and places relevant to the medieval coastal civilization, and it is important to recognize these biases. The geographical coverage is skewed towards Kenya, with the individuals from Tanzanian sites such as Songo Mnara, Kilwa and Lindi being sufficient to identify similarities and differences in ancestry profiles from Kenya but not sufficient to define a general pattern. In addition, the individuals that we analysed were not fully representative of all social and economic groups in Swahili society. Nearly all of the coastal graves and tombs in this study occupied prominent positions in medieval and early modern townscapes (see Supplementary

Information). With the possible exceptions of Kilwa, Lindi and Faza, we analysed elite individuals from high-profile coastal sites. However, the Swahili cultural world included many non-elite settlements, where ancestry might be systematically different[36]. Our analysis of data from two samplings of present-day people who identify as Swahili with different strategies for determining this identity also reveals qualitative differences in ancestry patterns, revealing how groups identified as Swahili retain high substructure and variation today.

These findings highlight multiple directions for future work on ancient DNA. One approach is to study individuals pre-dating the twelfth century, including before and after the major population mixtures that we show occurred around AD 1000. Another approach is to study individuals from unsampled parts of the Swahili world, including the present-day countries of Somalia, Mozambique, the Comoros Islands and Madagascar. However, the results presented here provide unambiguous evidence of ongoing cultural mixing on the East African coast for more than a millennium, in which African people interacted and had families with immigrants from other parts of Africa and the Indian Ocean world. Narratives of ancestry on the eastern African coast have a complex history, and the genetic findings of long-standing, sex-biased mixtures add to this complexity.

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

## Methods

### Inclusion and ethics

The present-day communities of the Swahili coast have strong traditions of connection to the people of the medieval coastal towns—including in some cases a tradition of descent from the people who lived in these communities, as well as shared language and religion—and thus community consultation is an important part of this work. Many medieval and present-day coastal people are also Muslims, and thus it is important to carry out any analysis in ways that are sensitive to Muslim proscriptions against disturbing the dead. The sampling for this study emerged from decades-long community-based archaeology projects—including by S.K. and C.M.K. in the Kenya region, and by S.W.-J., J.F., Tanzanian Antiquities and the Songo Mnara Ruins Committee in the Tanzania region—that involved participation in and return of results to local communities, following rules for handling and reburying of remains agreed by the communities. Prior to submission of this study for publication, the corresponding authors held return-of-results consultation meetings in Lamu, Songo Mnara and Kilwa Kisiwani, and feedback from these engagements was incorporated into the final manuscript. The community meeting held in Lamu was both in person and on zoom. Mtwapa no longer has a viable Swahili community. The descendants of the residents of Manda are no longer visible, but given Manda's proximity to Lamu, we chose Lamu as the most viable community to host the forum. Makwasinyi elders were consulted in 2001 and 2002 during fieldwork and again to discuss the results of the present analysis.

### Ancient DNA data generation

To generate the genetic data for the samples in Extended Data Table 1 and Supplementary Data File 2 (by-library metrics in Supplementary Data File 1), we used established protocols in dedicated clean rooms, involving first sampling typically 40 mg of powder from skeletal remains, then extracting DNA using methods designed to retain short, damaged fragments[37], then building individually barcoded libraries[38] after incubation with uracil DNA glycosylase (UDG) with the goal of greatly reducing the characteristic errors typical of ancient DNA. We amplified libraries and carried out in-solution enrichment of them for about 1.2 million SNPs[39] as well as baits targeting the mitochondrial genome[40]. We sequenced the enriched products along with a small amount of unenriched library on Illumina NextSeq 500 or HiSeqX10 instruments. The resulting sequenced paired-end reads were separated into respective libraries and stripped of identification tags and adapter related sequences. Read pairs were merged prior to alignment, requiring a minimum of 15 base pair overlap, allowing one mismatch if base quality was at least 20, or up to three mismatches at lower base quality. We restricted to sequences that were at least 30 base pairs long. The resulting sequences were aligned using samse from bwa-v0.6.1[41] separately to both the human genome reference sequence (hg19) and the mitochondrial RSRS genome[42] to allow the targeted nuclear and mitochondrial specific sequences to be assessed.

We performed a series of quality control measures. We inferred the rate of mismatch of the sequences mapping to the mitochondrial consensus sequence (of libraries with at least twofold coverage) and flagged libraries where the upper bound of the inferred match rate[43] was at least 95%; we also examined the rate of polymorphism on the X chromosome in males and flagged individuals as appreciably contaminated if the lower bound of the 95% confidence interval[44] was at least 1%. We tested if the ratio of Y chromosome to the sum of X and Y chromosome sequences was in the range expected for DNA entirely from a female (<3%) or male (>35%), and we tested for an elevated rate of cytosine to thymine substitution in the final nucleotide using a threshold of at least 1.8% (this was a lower threshold than is typical in ancient DNA studies due to the young age of the samples; the median damage rate of libraries is 9.5%). If a sample had sufficient evidence of contamination, we restricted to only damaged molecules of sequenced DNA, and these samples are denoted with '_d' in their Genetic ID in Supplementary Data File 2. Several samples did not pass quality control; these are any samples where all libraries failed in Supplementary Data File 1. No individual from two sampled sites—Bungule and Shanga—passed quality control. For each individual, we represented each position in the genome based on a single randomly selected sequence

A total of 80 individuals passed quality control screening. However, the analyses in this study focus on the 54 non-outlier individuals that are not first- or second-degree relatives of another individual in the study for which we have better quality data and for which we have data on a minimum of 15,000 SNPs. Relative relationships are determined by a method similar to READ[45].

Supplementary Data File 1 provides a full report of sequencing results on both libraries that yielded data passing standard measures of ancient DNA authenticity and libraries that failed screening; we also report negative results as a guide for future sampling efforts.

### Genotyping of modern Swahili individuals

We genotyped 93 present-day Swahili individuals, originally collected in ref. [21], on the Affymetrix Human Origins SNP array[26]. As described in the study where mtDNA and Y chromosome haplogroup information for these individuals was initially reported[21], individuals known to the research team to have origins in or near the targeted recruitment communities assisted in participant recruitment in a form of 'snowball sampling'. On sampling days, potential participants belonging to families known to have long residency in the community were approached and, if they consented to participation, would be asked afterwards if they knew of anyone else not closely related to them who might be available that day who met the inclusion criteria of being a healthy individual of at least 18 years of age who is a resident of that town and whose grandparents all identify as belonging to a group typically included in the broader Swahili identity. Each participant was provided information about the project in either English or Swahili from Institutional Review Board (IRB)-approved forms and those who elected to continue provided written consent. Protocols for collecting saliva and genealogical information were reviewed and approved by the Lehman College (no. 141-09-070) and City University of New York Integrated IRB (no. 323935). The saliva samples and genealogical information were collected in December 2009 and January 2010 in the Kenyan towns of Faza, Jomvu Kuu, Kizingitini, Lamu, Mikindani, Ndau, Pate, Siyu, Tchundwa and Wasini. A total of 96 samples were chosen for whole-genome analysis to maximize geographic spread, with 16 individuals each from Faza, Kizingitini, Ndau, Pate, Wasini and Jomvu Kuu. We limited to individuals who self-reported that the origin of both their parents and grandparents are local and Swahili, with a priority for males. Data from a total of 93 individuals passed quality control and their data are reported and analysed.

We genotyped on the Human Origins SNP array 19 newly reported individuals from Madagascar, for whom samples were collected between 2007–2014 with approval by the Human Subjects' Ethics Committees of the Health Ministry of Madagascar and by committees in France (Ministry of Research, National Commission for Data Protection and Liberties and Persons Protection Committee). Individuals all gave written consent before the study. Samples came from two locations in the north and south of the country. Sampled villages were founded before 1900, and individuals were 61 ± 15 years old, with the maternal grandmother and paternal grandfather born within 50 km of the sampling location.

We genotyped on the Human Origins SNP array 10 newly reported Emirati samples that were collected from Emirati nationals from the city of Al Ain, United Arab Emirates. Ethical approval was obtained from the Al Ain District Human Research Ethical Committee. The samples were collected from healthy adults, and first genotyped on another SNP array as controls for a rare disease study[46].

A list of the newly reported individuals with data on the Human Origins SNP array and some relevant information can be found in Supplementary Data File 4.

## Principal component analysis

We used smartpca version 18180 from EIGENSOFT version 8.0.0[47] with optional parameters (numoutlieriter: 2, numoutevec: 3, lsqproject: YES, newshrink: YES, and hiprecision: YES). We computed eigenvectors of the covariance matrix of SNPs from present-day individuals from Africa and Eurasia genotyped on the Affymetrix Human Origins (HO) SNP array, which targets approximately 600,000 SNPs that are a subset of those targeted by in-solution enrichment (see Supplementary Information for a list of populations used). Ancient individuals and present-day individuals (most genotyped on the HO SNP array, and some also genotyped on the Illumina Human Omni5 Bead Chip[32]) were projected into the three-dimensional (3D) eigenspace determined by the first three principal components.

## ADMIXTURE clustering analysis

We prepared data for the ADMIXTURE[48] plots in Fig. 1c and Extended Data Fig. 1 using PLINK2[49]. We applied the maf 0.01 option, which only includes SNPs with minor allele frequency of at least 0.01. We pruned SNPs based on LD, using the indep-pairwise option with a window size of 200 variants, a step size of 25 variants, and a pairwise $r^2$ threshold of 0.4. We ran 4 replicates of ADMIXTURE with random seeds with $K = 4$ to $K = 12$ ancestral reference populations.

## Estimates of mixture proportions

We used qpAdm (ADMIXTOOLS version 7.0.2)[26] to test if the Makwasinyi, Manda, Mtwapa and Faza, Kilwa, and Songo Mnara target populations can be formally modelled as having derived from a set of source populations (termed 'left' groups) relative to a set of reference populations (termed 'right' groups). See Supplementary Information for further details. qpAdm also provides a $P$ value for fit to the model based on a block jackknife across all 5 cM segments of the autosomes.

In our application of qpAdm, we use a cycling approach, treating the target as a linear combination of all possible subsets of the candidate source populations, and moving the other candidate source population to the right. Cycling populations to the right allows us to test if a proposed set of left source populations is consistent with being more closely related to the target than other populations. Thus, we can build the closest admixture model within the constraints of our dataset and test its fit to the data.

## Y and mitochondrial haplogroups

Mitochondrial haplogroups were determined using HaploGrep 2[50]. Y chromosome haplogroups were determined according to the Yfull tree version 8.09 (https://github.com/YFullTeam/YTree/blob/master/ytree/tree_8.09.0.json)[51].

## Sex bias from X-autosome comparisons

We estimated the extent to which the ancestry from each source population was contributed by female and male ancestors. To do this, we compared the inferences of proportions of ancestry on chromosomes 1–22 (the autosomes) which reflect 50% female and 50% male inheritance (the autosomal coefficient for proportion of a specific ancestry, $C_A = (m + f)/2$), and on chromosome X, which reflects 67% female and 33% male ancestry (the X chromosome coefficient for proportion of a specific ancestry, $C_X = (2f + m)/3$). We determined $Z$-scores for differences between these two estimates as in ref. [52]. For each of Mtwapa (including the one Faza individual), Manda, Kilwa and Makwasinyi, we quantified sex bias in the following manner. (1) We sampled $10^6$ sets of coefficients by generating random numbers from a multivariate normal distribution based on the qpAdm-determined jackknife mean coefficients and the error covariance matrix for both the autosomes and

chromosome X, separately. We ensured that the eigenvalues of the matrices are all greater than or equal to zero by adding a small offset to the matrices determined as the absolute value of its respective minimum eigenvalue. (2) We removed from consideration all infeasible sets of coefficients, namely those that include a coefficient below 0. (3) We next calculated the female and male proportions of ancestry for each source population. Given the coefficient $C_A$ on the autosomes and the coefficient $C_X$ on the X chromosome, we can determine the likely proportion of female and male ancestry from any given source by solving the system of two above equations, giving $f = 3C_X − 2C_A$ and $m = 4C_A − 3C_X$. We calculate these $f$ and $m$ proportions for each source population for the sampled set of autosomal and X chromosome coefficients. (4) We normalize the proportions of female and male ancestry as $\hat{f} = f/(f + m)$ and $\hat{m} = m/(f + m)$ for each source for the sampled sets. (5) We found the mean and standard deviation of all the $\hat{f}$ and $\hat{m}$ values from the sampled set for each of the source populations. (3) We took 95% confidence intervals of all the sampled sets of ancestry proportions and record the values as ranges in Table 1. Under the simple model of all the mixture occurring at once, these formulae can be interpreted as estimating the fraction of ancestors at the time of mixing that are on the male or female side. However, if the mixture was more gradual, the interpretation is more complicated albeit still informative about sex bias.

## Proximate sources of the coastal cline

We plot all the Mtwapa, Faza and Manda individuals on a ternary plot with their respective proportions of Makwasinyi, Iranian and Indian ancestry. We apply a linear regression to the high coverage individuals from Mtwapa (I19381, I19394, I19384, I19414, I19420, I19413, I19417, I19401, I23662, I17409, I19391, I13611, I17412, I17413, I23558, I19415, I23561, and I21475) and Manda (I7934) as seen in the ternary plot (see Fig. 2b and Supplementary Fig. 3). The associated Cartesian equation is $f(x) = 0.09856x + 0.01301$. The line intersects the Makwasinyi axis at 98 ± 6% (left circle in Fig. 2b and Supplementary Fig. 3), allowing for 2 ± 6% Indian ancestry. This is statistically consistent with 100% Makwasinyi and 0% Iranian and 0% Indian ancestries. The other end of the regression line intersects the Indian axis (right circle in Fig. 2b and Supplementary Fig. 3) at 12 ± 5% Indian ancestry and 88 ± 5% Iranian ancestry. We further refined these proportions by jackknifing the coefficient estimates. We estimate that one admixing source population is 98.4 ± 2.5% Makwasinyi-related, consistent with a parsimonious model of 100% Makwasinyi ancestry at that end of the cline. The other source population is 12.2 ± 2.8% Indian and 87.8 ± 2.8% Iranian (see Supplementary Information).

## Determining the date of mixture

DATES measures the covariance between pairs of positions in the genome separated by a specified genetic distance, for an admixture model with two source populations[30,31]. This analysis is applied separately to all individuals, and the inferences are pooled across individuals to increase resolution. We pool the individuals from Mtwapa, Faza, and Manda into a northern group, and the individuals from Kilwa and Songo Mnara I19550 into a southern group. We use a 28 ± 2 year-to-generation conversion estimate[53] to calculate the average date of the admixture.

## Reporting summary

Further information on research design is available in the Nature Portfolio Reporting Summary linked to this article.

## Data availability

Bam files of aligned reads for the 80 newly published ancient individuals can be obtained from the European Nucleotide Archive (accession no. PRJEB58698). SNP array genotype data for 122 newly reported modern individuals can be obtained from a permanent link at the Dataverse repository at (https://doi.org/10.7910/DVN/NC28XW). The data for

103 of these individuals can be downloaded without registering, whereas the informed consent for 19 individuals from Madagascar is not consistent with unmediated public posting of data, and may be downloaded after filling out a form with an email address and institutional or professional affiliation, and including an affirmation of the following statements: (a) I will not distribute the data outside my collaboration; (b) I will not post the data publicly; (c) I will make no attempt to connect the genetic data to personal identifiers for the samples; (d) I will use the data only for studies of population history; and (e) I will not use the data for commercial purposes.

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

**Acknowledgements** We thank N. Adamski, R. Bernardos, B. Culleton, I. Greenslade, D. Kennett, M. Mah, A. Micco and Z. Zhang for contributions to data generation, processing and curation. E.S.B. was supported by EMBO Postdoctoral fellowship ALTF 242-2021. The work of C.M.K., S.B.K., S.R.W. and J.M. in Kenya was supported by the National Museums of Kenya and the Republic of Kenya. The bulk of the research at Manda, Mtwapa, Faza and Makwasinyi was carried out when C.M.K. was at the Field Museum of Natural History. We acknowledge financial support from the US National Science Foundation SBR 9024683 (1991–1993); BCS 9615291 (1996–1998); BCS 0106664 (2002–2004); BCS 0352681 (2003–2004); BCS 0648762 (2007–2009; BCS 1030081 (2010–2012), the US National Endowment for the Humanities (2012–2014), the US IIE J. W. Fulbright Sr Scholars Program 2002–2003; 2012), and the National Geographic Society (1996–1997). The research of S.W.-J. and J.F. at Songo Mnara was supported by the Antiquities Division of the Ministry of Natural Resources and Tourism, Tanzania, and also by the National Science Foundation (BCS 1123091) and the Arts and Humanities Research Council (AH/J502716/1). The Madagascar modern sample collection was supported by the MAGE consortium. D.R. is an Investigator of the Howard Hughes Medical Institute, and the ancient DNA laboratory work and analysis were also supported by National Institutes of Health grant HG012287, by John Templeton Foundation grant 61220, by the Allen Discovery Center program, which is a Paul G. Allen Frontiers Group advised program of the Paul G. Allen Family Foundation, and by a gift from J.-F. Clin. S.W.-J. and all co-authors are grateful to UK Research and Innovation for enabling fully Open Access publication of this study via financial support to the York Open Access Fund.

**Author contributions** E.S.B., J.F., S.W.-J., S.R.W., J.M., M.E.P., D.R. and C.M.K. conceptualized the study. E.S.B., K.S. and S.M. formally analysed the genetic data. E.S.B., N.B., K.C., E.C., L.I., A.M.L., J.O., L.Q., K.S., J.N.W., F.Z., B.J.C., S.M., N.R., N.P., K.B.A., M.M.M., S.K. and D.R. were involved in the genetic data investigation. J.F., S.W.-J., J.O., G.A., A.O.G., A. Kabiru, A. Kwekason, A.Z.P.M., F.K.M., E.N., C.O., E.S., L.A.-G., B.R.A., S.B.-S., T.L., D.P., C.R., J.-A.R., R.L.R., B.J.C., M.A.M., K.B.A., M.M.M., S.R.W., J.M., S.K., M.E.P., D.R. and C.M.K. provided resources. E.S.B., J.F., S.W.-J., M.E.P., D.R. and C.M.K. wrote the paper.

**Competing interests** The authors declare no competing interests.

**Additional information**
**Correspondence and requests for materials** should be addressed to Esther S. Brielle, Jeffrey Fleisher, Stephanie Wynne-Jones, David Reich or Chapurukha M. Kusimba.

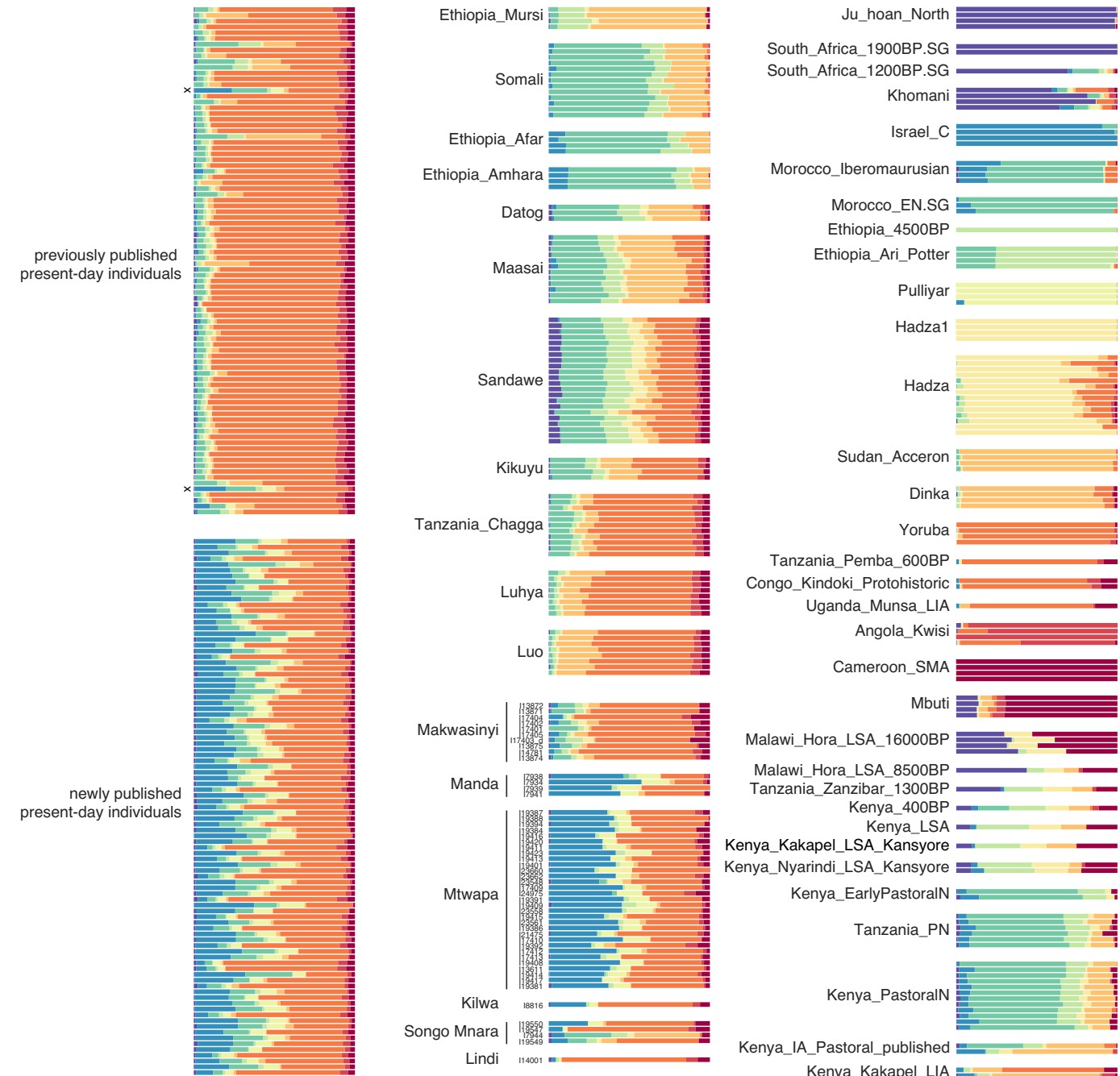

**Extended Data Fig. 1 | ADMIXTURE analysis.** We analyzed 6422 individuals including African populations genotyped with the Human Origins SNP array, along with 1240K capture ancient individuals from present-day Kenya, present-day Tanzania, and surrounding areas; Swahili coast present-day samples;[21,32] ancient Israel_Natufian; and Pulliyar samples from present-day India. We chose K = 10 ancestral reference populations (chosen based on low cross-validation errors, high log-likelihood scores, and a low number of reference populations so as to not overfit) to highlight various ancestries found throughout Eastern Africa). We were able to discern a variety of African pastoral, farming, and forager ancestries within coastal and inland Swahili-speaking populations of the Human Origins dataset. The Makwasinyi reference ancestry composition resemble those of the present-day coastal populations sampled in a previously published dataset[32], albeit with different proportions. The x's identify two outlier individuals from the dataset with more Asian admixture; we removed them for population-level modeling. The genome-wide ancestry composition of the ancient coastal samples more closely resembles that of the newly published present-day individuals for which haplogroup data was published in[21].

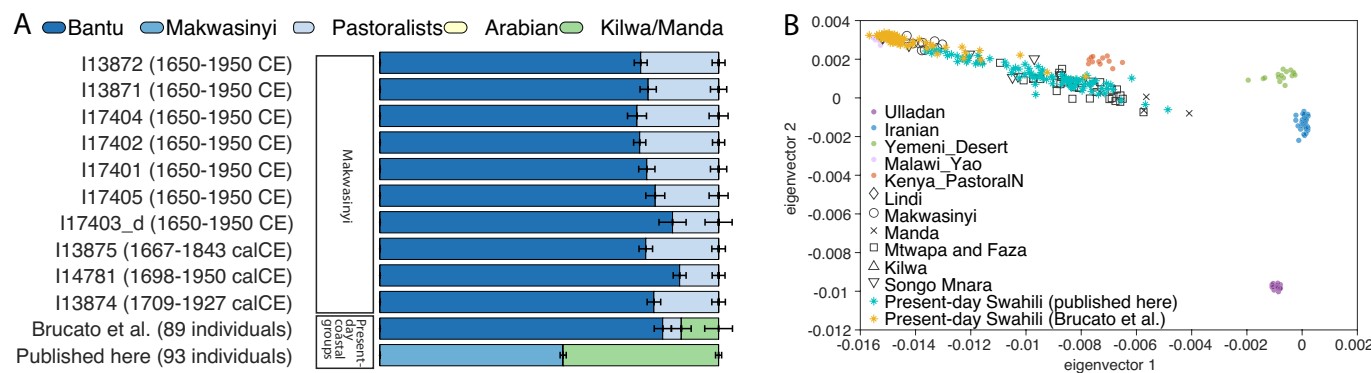

**Extended Data Fig. 2 | Relationship of medieval to recent East Africans.**
A) Graphical representation of individual and population level *qpAdm* models
for Makwasinyi individuals, 89 present-day people who identify as and speak
Swahili[32], and 93 present-day people who identify as Swahili and have long
familial residency in the town, sampled from 6 coastal regions along Kenya[21].
The green ancestry component represents the modeled contribution from
medieval Swahili coast people, as proxied by individuals buried in Kilwa or
Manda. One standard error bars around the mean are computed using a
Block Jackknife across all 5 centimorgan segments of the autosomes, and are
meaningful even for single individuals as the genome reflects ancestral
contributions of large numbers of ancestors. B) PCA with the same present-day
Swahili-identifying people (asterisks) and other present and ancient people
(filled circles)[23,54,55], and ancient individuals from eastern Africa published in
this study (open shapes)[23,24] projected onto the first two eigenvectors.

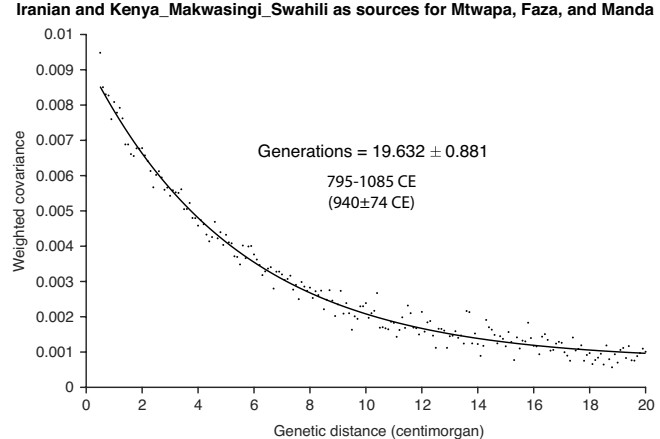

**Iranian and Kenya_Makwasingi_Swahili as sources for Mtwapa, Faza, and Manda**

Generations = 19.632 ± 0.881

795-1085 CE
(940±74 CE)

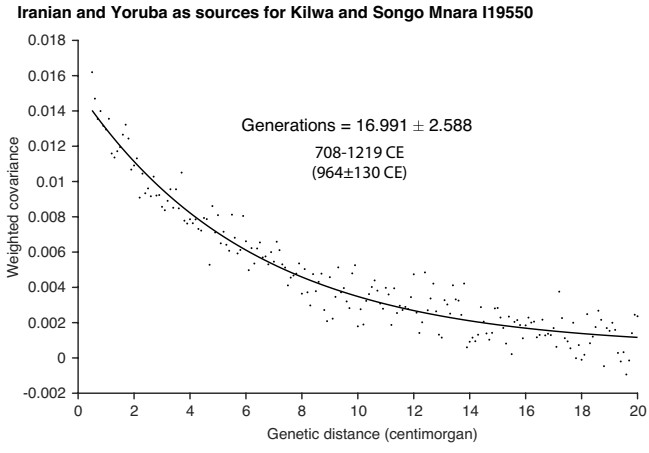

**Iranian and Yoruba as sources for Kilwa and Songo Mnara I19550**

Generations = 16.991 ± 2.588

708-1219 CE
(964±130 CE)

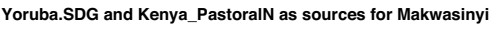

**Yoruba.SDG and Kenya_PastoralN as sources for Makwasinyi**

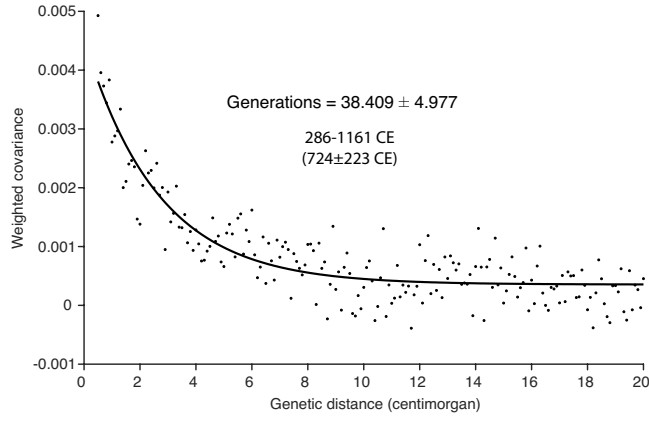

Generations = 38.409 ± 4.977

286-1161 CE
(724±223 CE)

**Makwasinyi and Iranian as sources for newly published present-day Swahili**

Generations = 27.221 ± 0.917

1096-1410 CE
(1253±80 CE)

**Extended Data Fig. 3 | Estimates of dates of population mixture.** Curves that show the exponential decay of linkage disequilibrium generated by mixture between two populations related differentially to the two sources as a function of the number of elapsed generations since the mixture event. Multiplying by a 28 ± 2 year-to-generation conversion factor[53] and subtracting from the average calibrated radiocarbon dates or the archaeologically estimated date of the ancient individuals gives estimates in calendar years. For present-day Swahili individuals, we subtract from the year 2015, when sampling occured.

## Extended Data Table 1 | Ancient individuals with high coverage genome-wide data newly reported in this study

| ID | Date | Locality | SNPs | Sex | Y haplogroup | mtDNA haplogroup | Comments |
|---|---|---|---|---|---|---|---|
| I14781 | 1698-1950 calCE | Taita Taveta, Makwasinyi | 532619 | F | ---- | L0a1a+200 | |
| I13872 | | Taita Taveta, Makwasinyi | 598510 | M | E1b1a1a1a2a1a | L4b2a | Family A |
| I13875 | 1667-1843 calCE | Taita Taveta, Makwasinyi | 537980 | M | E1b1a1a1a2a1a3b1d1c | L2a1+143 | Family A |
| I13874 | 1709-1927 calCE | Taita Taveta, Makwasinyi | 446807 | M | E1b1a1a1a2a1a3a1d~ | L3a2 | |
| I13871 | | Taita Taveta, Makwasinyi | 181981 | F | ---- | L3d1a1a | |
| I13873 | | Taita Taveta, Makwasinyi | 240344 | M | E1b1a1a1a2a1a | L2a1+143 | Removed family A |
| I17404 | | Taita Taveta, Makwasinyi | 93611 | F | ---- | L3d1a1a | |
| I17402 | | Taita Taveta, Makwasinyi | 593614 | M | E1b1a1a1a2a1a3b1d1c | L1c3b1a | Family A |
| I17401 | | Taita Taveta, Makwasinyi | 228632 | F | ---- | L3f2a1 | |
| I17405 | | Taita Taveta, Makwasinyi | 110661 | F | | L0f2a | |
| I17403 | | Taita Taveta, Makwasinyi | 36275 | F | | L5a1a | |
| I7934 | 1456-1621 calCE | Lamu, Manda \| Kenya | 403873 | F | ---- | L2d1a | Family A |
| I7939 | 1456-1621 calCE | Lamu, Manda \| Kenya | 44918 | M | J2 | n/a (<2x) | |
| I7941 | 1485-1629 calCE | Lamu, Manda \| Kenya | 44105 | M | G2a | L3e3a | |
| I7942 | 1457-1626 calCE | Lamu, Manda \| Kenya | 391682 | M | J2b2a2~ | L2d1a | Removed family A |
| I19407 | | Lamu, Manda \| Kenya | 25216 | M | J2 | L2d1a | Removed family A |
| I7938 | 1437-1482 calCE | Lamu, Manda \| Kenya | 71503 | F | ---- | L2a1b1 | |
| I17410 | | Kilifi, Mtwapa \|Kenya | 144148 | M | J1a2a1a2d2b~ | L3d1a1a | |
| I17409 | | Kilifi, Mtwapa \|Kenya | 636684 | M | J1a2a1a2d2b~ | M30d1 | Family B |
| I19381 | | Kilifi, Mtwapa \|Kenya | 367125 | M | J1a2a1a2d2b~ | L2a1a | |
| I19387 | | Kilifi, Mtwapa \|Kenya | 183657 | F | ---- | L2a1b1a | |
| I19388 | | Kilifi, Mtwapa \|Kenya | 26134 | M | E1b1b | L0a2a2a | Family B |
| I19394 | | Kilifi, Mtwapa \|Kenya | 202755 | M | E1b1b1b2a1a1a1a1f~ | L0a2a2a | Family B |
| I19390 | | Kilifi, Mtwapa \|Kenya | 19762 | F | | L3b1a+@16124 | outlier |
| I19392 | | Kilifi, Mtwapa \|Kenya | 134037 | F | ---- | L3b1a1 | |
| I19384 | | Kilifi, Mtwapa \|Kenya | 463984 | F | | L3b1a1a | Family C |
| I19414 | | Kilifi, Mtwapa \|Kenya | 902017 | F | | L1c3a1b | Family E |
| I19419 | | Kilifi, Mtwapa \|Kenya | 126541 | M | J1a2a1a2 | L3f1b1a1 | Removed family C |
| I19416 | | Kilifi, Mtwapa \|Kenya | 121558 | M | R1a1a1 | L3b1a1a | Family D |
| I19420 | 1446-1614 calCE | Kilifi, Mtwapa \|Kenya | 193510 | M | R1a1a1 | L3e1d1 | |
| I19411 | 1496-1630 calCE | Kilifi, Mtwapa \|Kenya | 163986 | F | ---- | L1c3a1b | |
| I19423 | | Kilifi, Mtwapa \|Kenya | 56327 | M | E1b1b | L0d3 | Family B |
| I19408 | | Kilifi, Mtwapa \|Kenya | 33031 | M | J1a2a1a2d2b~ | L3e3a | |
| I19413 | | Kilifi, Mtwapa \|Kenya | 725990 | M | J1a2a1a2d2b~ | L1b1a | |
| I19417 | | Kilifi, Mtwapa \|Kenya | 193918 | M | J1a2a1a2 | L3e3a | Family D |
| I19401 | | Kilifi, Mtwapa \|Kenya | 765624 | M | J1a2a1a2d2b~ | L0a2a2a | Family A |
| I23660 | | Kilifi, Mtwapa \|Kenya | 38908 | M | J | L3d1a | |
| I23662 | | Kilifi, Mtwapa \|Kenya | 214343 | M | J2a1a1a2a2b | L0a2a1a2 | Family F |
| I23550 | | Kilifi, Mtwapa \|Kenya | 20760 | F | | M30d1 | Removed family B |
| I24975 | | Kilifi, Mtwapa \|Kenya | 19369 | F | ---- | n/a (<2x) | |
| I19391 | | Kilifi, Mtwapa \|Kenya | 222944 | M | J1a2a1a2d2b~ | L0a2a2a | Family B |
| I19409 | | Kilifi, Mtwapa \|Kenya | 92063 | M | J1 | L0a1b1a | |
| I13611 | | Kilifi, Mtwapa \|Kenya | 525800 | F | ---- | L3b1a11 | Family A |
| I17412 | | Kilifi, Mtwapa \|Kenya | 973533 | M | J1a2a1a2d2b~ | L0a2a1a2 | |
| I17413 | | Kilifi, Mtwapa \|Kenya | 636350 | F | ---- | L0a1b2a | |
| I19386 | | Kilifi, Mtwapa \|Kenya | 80546 | M | J1 | L3e3a | |
| I19415 | | Kilifi, Mtwapa \|Kenya | 518120 | F | ---- | L0a2a1a2 | Family E |
| I23548 | | Lamu, Pate Island, Faza | 132420 | M | E1b1a1a1a2a1a | L3e3a | |
| I23558 | | Mtwapa \| Kenya | 699608 | F | | L0a1b1a | Family B |
| I21475 | | Mtwapa \| Kenya | 928732 | F | ---- | L3f1b1a1 | Family C |
| I23561 | | Mtwapa \| Kenya | 753944 | F | | L3a2 | |
| I7944 | 1516-1667 calCE | Songo Mnara \| Tanzania | 633887 | M | E1b1b1b2b2a1a~ | L3a1b | |
| I19551 | 1294-1392 calCE | Songo Mnara \| Tanzania | 19083 | M | R1a | L3e3a | |
| I19548 | 1418-1450 calCE | Songo Mnara \| Tanzania | 79001 | M | E1b1a1~ | L3e2b | |
| I19552 | 1402-1437 calCE | Songo Mnara \| Tanzania | 31712 | F | | L3e2b1a2 | |
| I19547 | 1508-1648 calCE | Songo Mnara \| Tanzania | 472669 | F | ---- | L3d1a1a1 | Family A |
| I19550 | 1412-1446 calCE | Songo Mnara \| Tanzania | 66333 | F | | L3d1a1a | |
| I19549 | 1629-1794 calCE | Songo Mnara \| Tanzania | 248300 | M | E1b1b1a1a1b2~ | L3d1a1a | Family A |
| I8816 | 1300-1600 CE | Kilwa, Coast \| Tanzania | 880458 | M | J2a2a1a1a2a~ | L2a1h | |
| I14001 | 1511-1664 calCE | Lindi \| Tanzania | 810172 | M | E1b1a1a1a2a1a3a1d~ | L0a1a2 | |

The lower coverage individual (but still above 15,000 SNPs) from pairs of first-degree relatives are not included in whole-genome analyses, but are reported here. Makwasinyi Family A has 4 members: I13873 (removed from analysis) has a son I13872 and a brother I13875 (who is genetically a second or third-degree relative of I13872), and all three males are second- or third-degree relatives of I17402. Manda Family A has 4 members: I7934, I7942 (removed from analysis), I7943 (not included in the table due to low sequencing coverage), and I19407 (removed from analysis) are all first-degree relatives of one another. Mtwapa Family A has 2 members who are second- or third-degree relatives, I19401 and I13611. Mtwapa Family B has 8 members; of them, I23550 and I17409 are first-degree relatives. Second- or third-degree relatives are I17409 to I19391, I19394 to I23558, I19394 to I19391, I19391 to I19393 (not included in the table due to low sequencing coverage), I19394 to I19423, and I19394 to I19388. Mtwapa Family C has 3 members: I21475 and I19419 are first-degree relatives, and I21475 and I19384 are second- or third-degree relatives. Mtwapa Family D has 2 members who are second- or third-degree relatives, I19416 and I19417. Mtwapa Family E has 2 members who are second- or third-degree relatives, I19414 and I19415. Mtwapa Family F has 2 members who are second- or third-degree relatives, I23662 and I19398 (not included in the table due to low sequencing coverage, this individual also has a direct date included in Supplementary Data File 2). Songo Mnara Family A has 2 members who are second or third-degree relatives, I19547 and I19549.

**Extended Data Table 2 | Ancient individuals with high coverage genome-wide data newly reported in this study that have significant evidence of Persian and often Indian ancestry**

| Date | ID | Swahili group | Local African source | P-value | African ancestry | Persian ancestry | Indian ancestry |
|---|---|---|---|---|---|---|---|
| 1300-1600 CE | I8816 | Kilwa | Lindi (Bantu proxy) | 0.2742 | 0.76±0.01 | 0.24±0.01 | -- |
| 1412-1446 calCE | I19550 | Songo Mnara | Lindi (Bantu proxy) | 0.3758 | 0.67±0.02 | 0.25±0.04 | 0.08±0.04 |
| 1437-1482 calCE | I7938 | | Makwasinyi | 0.4391 | 0.42±0.02 | 0.54±0.03 | 0.04±0.03 |
| 1456-1621 calCE | I7934 | Manda | Makwasinyi | 0.7001 | 0.3±0.01 | 0.63±0.02 | 0.08±0.02 |
| 1456-1621 calCE | I7939 | | Makwasinyi | 0.3931 | 0.57±0.02 | 0.38±0.04 | 0.05±0.04 |
| 1485-1629 calCE | I7941 | | Makwasinyi | 0.6402 | 0.46±0.02 | 0.42±0.04 | 0.12±0.04 |
| 1250-1650 CE | I19387 | | Makwasinyi | 0.2390 | 0.59±0.02 | 0.34±0.03 | 0.07±0.02 |
| 1200-1450 CE | I19388 | | Makwasinyi | 0.7668 | 0.55±0.03 | 0.41±0.05 | 0.05±0.05 |
| 1200-1450 CE | I19394 | | Makwasinyi | 0.0020 | 0.62±0.02 | 0.37±0.02 | 0.01±0.02 |
| 1250-1650 CE | I19384 | | Makwasinyi | 0.0547 | 0.5±0.02 | 0.42±0.02 | 0.08±0.02 |
| 1445-1609 calCE | I19416 | | Makwasinyi | 0.8258 | 0.7±0.02 | 0.23±0.03 | 0.07±0.02 |
| 1446-1614 calCE | I19420 | | Makwasinyi | 0.3127 | 0.66±0.02 | 0.26±0.02 | 0.08±0.02 |
| 1496-1630 calCE | I19411 | | Makwasinyi | 0.2720 | 0.61±0.02 | 0.29±0.03 | 0.1±0.02 |
| 1323-1423 calCE | I19423 | | Makwasinyi | 0.7124 | 0.46±0.02 | 0.48±0.04 | 0.05±0.04 |
| 1250-1650 CE | I19413 | | Makwasinyi | 0.4221 | 0.57±0.01 | 0.36±0.02 | 0.06±0.02 |
| 1250-1650 CE | I19401 | | Makwasinyi | 0.0468 | 0.57±0.01 | 0.39±0.02 | 0.04±0.02 |
| 1250-1650 CE | I23660 | | Makwasinyi | 0.1056 | 0.62±0.03 | 0.34±0.04 | 0.04±0.04 |
| 1450-1700 CE | I23662 | | Makwasinyi | 0.0024 | 0.53±0.02 | 0.41±0.02 | 0.05±0.02 |
| 1500-1700 CE | I23548 | | Makwasinyi | 0.7622 | 0.64±0.02 | 0.3±0.03 | 0.06±0.03 |
| 1226-1297 calCE | I17409 | | Makwasinyi | 0.4021 | 0.5±0.01 | 0.43±0.02 | 0.07±0.02 |
| 1250-1650 CE | I24975 | Mtwapa and Faza | Makwasinyi | 0.4748 | 0.47±0.03 | 0.42±0.06 | 0.1±0.06 |
| 1200-1450 CE | I19391 | | Makwasinyi | 0.2653 | 0.48±0.02 | 0.44±0.02 | 0.08±0.02 |
| 1442-1612 calCE | I19409 | | Makwasinyi | 0.2347 | 0.45±0.02 | 0.47±0.03 | 0.08±0.03 |
| 1200-1450 CE | I23558 | | Makwasinyi | 0.0003 | 0.42±0.01 | 0.48±0.02 | 0.1±0.02 |
| 1408-1442 calCE | I19415 | | Makwasinyi | 0.7634 | 0.63±0.01 | 0.32±0.02 | 0.05±0.02 |
| 1424-1457 calCE | I23561 | | Makwasinyi | 0.2154 | 0.5±0.01 | 0.45±0.02 | 0.05±0.02 |
| 1451-1619 calCE | I19386 | | Makwasinyi | 0.4199 | 0.55±0.02 | 0.47±0.03 | -0.02±0.03 |
| 1250-1650 CE | I21475 | | Makwasinyi | 0.2545 | 0.71±0.01 | 0.27±0.02 | 0.03±0.01 |
| 1250-1650 CE | I17410 | | Makwasinyi | 0.7735 | 0.44±0.02 | 0.46±0.03 | 0.1±0.02 |
| 1250-1650 CE | I19392 | | Makwasinyi | 0.2057 | 0.76±0.02 | 0.19±0.03 | 0.05±0.02 |
| 1250-1650 CE | I17412 | | Makwasinyi | 0.0616 | 0.49±0.01 | 0.41±0.02 | 0.11±0.02 |
| 1250-1650 CE | I17413 | | Makwasinyi | 0.0616 | 0.49±0.01 | 0.41±0.02 | 0.11±0.02 |
| 1250-1650 CE | I19408 | | Makwasinyi | 0.7674 | 0.54±0.03 | 0.41±0.05 | 0.05±0.05 |
| 1250-1650 CE | I13611 | | Makwasinyi | 0.5060 | 0.59±0.02 | 0.36±0.02 | 0.05±0.02 |
| 1350-1500 CE | I19414 | | Makwasinyi | 0.2761 | 0.61±0.01 | 0.32±0.02 | 0.07±0.02 |
| 1400-1650 CE | I19417 | | Makwasinyi | 0.4274 | 0.65±0.02 | 0.31±0.02 | 0.04±0.02 |
| 1250-1650 CE | I19381 | | Makwasinyi | 0.0255 | 0.6±0.02 | 0.34±0.02 | 0.06±0.02 |

The African proxy source for the individuals at Mtwapa, Faza, and Manda in Kenya is the Makwasinyi group newly reported in this study, and for the Kilwa individual and the Songo Mnara individual I19550 in Tanzania is the Lindi individual newly reported in this study. The Persian proxy source in all cases is the Human Origins genotyped Iranian group[54,55]. The Indian proxy source in all cases is the Human Origins genotyped Sahariya_MP group[56]. Individuals for whom the *qpAdm* model does not fit with statistical confidence (P < 0.01 by a Hotelling T-squared test) are shaded in gray, and one standard error bars around the mean are based on a Block Jackknife across the autosomes with 5 centimorgan blocks.

**Extended Data Table 3 | Individuals with high coverage genome-wide data that do not have evidence of Persian ancestry**

| Date | ID | Individual | P value | Forager | Malagasy | Bantu | Makwasinyi | Pastoral Neolithic | Pastoral Iron Age | Arabian | Medieval Swahili |
|---|---|---|---|---|---|---|---|---|---|---|---|
| 1511-1664 calCE | I14001 | Lindi | 0.5996 | 0.11±0.03 | | 0.89±0.03 | | | | | |
| 1508-1648 calCE | I19547 | Songo Mnara | 0.1261 | | 0.08±0.02 | 0.84±0.02 | | | | 0.08±0.01 | |
| 1516-1667 calCE | I7944 | Songo Mnara | 0.9535 | | | | | 0.03±0.07 | 0.98±0.07 | | |
| 1629-1794 calCE | I19549 | | 0.1421 | | | 0.79±0.01 | | | | 0.21±0.01 | |
| 1650-1950 CE | I13872 | Makwasinyi | 0.5336 | | | 0.77±0.02 | | 0.23±0.02 | | | |
| 1650-1950 CE | I13871 | | 0.6913 | | | 0.79±0.02 | | 0.21±0.02 | | | |
| 1650-1950 CE | I17404 | | 0.0138 | | | 0.76±0.03 | | 0.24±0.03 | | | |
| 1650-1950 CE | I17402 | | 0.3717 | | | 0.77±0.02 | | 0.23±0.02 | | | |
| 1650-1950 CE | I17401 | | 0.3161 | | | 0.79±0.02 | | 0.21±0.02 | | | |
| 1650-1950 CE | I17405 | | 0.1396 | | | 0.81±0.03 | | 0.19±0.03 | | | |
| 1650-1950 CE | I17403_d | | 0.3585 | | | 0.86±0.04 | | 0.14±0.04 | | | |
| 1667-1843 calCE | I13875 | | 0.3751 | | | 0.79±0.02 | | 0.22±0.02 | | | |
| 1698-1950 calCE | I14781 | | 0.4857 | | | 0.89±0.02 | | 0.12±0.02 | | | |
| 1709-1927 calCE | I13874 | | 0.3811 | | | 0.81±0.02 | | 0.19±0.02 | | | |
| present day | Pool of previously published individuals | Kenya coast | 0.4537 | | | 0.84±0.03 | | | 0.06±0.03 | | 0.11±0.04 |
| present day | Pool of newly published individuals | Kenya coast | 0.288 | | | | 0.541±0.009 | | | | 0.459±0.009 |

For the Lindi individual I14001 the forager proxy source is the Hadza group genotyped on the Human Origins array[54], and the Bantu proxy source is the Malawi_Yao group[23]. For the Songo Mnara individual I7944, the Pastoral Neolithic and Pastoral Iron Age proxy sources are the 1240K capture Kenya_PastoralN group[24] and the 1240K capture Kenya_IA_Pastoral_published group[24], respectively. For the Songo Mnara individual I19547 the Malagasy proxy source is the Madagascar_North group, the Bantu proxy source is the Malawi_Ngoni group, and the Arabian proxy source is the Yemeni_Highlands_Raymah individual. For Songo Mnara individual I19549 the Bantu proxy source is the 1240K capture Tanzania_Pemba_600BP_published individual[23], and the Arabian proxy source is the Emirati group. For the Makwasinyi group the Bantu proxy source is the 1240K capture Tanzania_Pemba_600BP_published individual[23], and the Pastoral Neolithic proxy source is the 1240K capture Kenya_PastoralN group[24]. The Bantu proxy source for all the present-day groups from Kilifi, Lamu, and Mombasa is the 1240K Lindi individual from the present study. For the previously published present-day group, the medieval Swahili-speaking proxy is the Kilwa individual, and the Pastoral Iron Age proxy source is two published individuals from[24], I8892_published and I12381. For the newly published present-day group, the medieval Swahili coast proxy source is the Manda group. *qpAdm* P-values are from a Hotelling T-squared test, and one standard error bars around the mean are based on a Block Jackknife across the autosomes with 5 centimorgan blocks.

**Extended Data Table 4 | Proportions of ancestry for present-day Swahili-identified individuals newly genotyped for this study**

| Swahili coastal proxy source with Eurasian ancestry | African proxy Source without Eurasian ancestry | P-value |
|---|---|---|
| 70.7±0.9% Mtwapa+Faza | 29.3±0.9% Makwasinyi | 0.024 |
| 76.5±0.8% Mtwapa+Faza | 23.5±0.8% Lindi | 0.024 |
| 45.9±0.9% Manda | 54.1±0.9% Makwasinyi | 0.29 |
| 52.8±0.9% Manda | 47.2±0.9% Lindi | 0.40 |

qpAdm P-values are from a Hotelling T-squared test, and one standard error bars around the mean are based on a Block Jackknife across the autosomes with 5 centimorgan blocks.

# Reporting Summary

## Statistics

For all statistical analyses, confirm that the following items are present in the figure legend, table legend, main text, or Methods section.

| n/a | Confirmed | |
|---|---|---|
| ☐ | ☒ | The exact sample size (*n*) for each experimental group/condition, given as a discrete number and unit of measurement |
| ☐ | ☒ | A statement on whether measurements were taken from distinct samples or whether the same sample was measured repeatedly |
| ☐ | ☒ | The statistical test(s) used AND whether they are one- or two-sided<br>*Only common tests should be described solely by name; describe more complex techniques in the Methods section.* |
| ☐ | ☒ | A description of all covariates tested |
| ☐ | ☒ | A description of any assumptions or corrections, such as tests of normality and adjustment for multiple comparisons |
| ☐ | ☒ | A full description of the statistical parameters including central tendency (e.g. means) or other basic estimates (e.g. regression coefficient) AND variation (e.g. standard deviation) or associated estimates of uncertainty (e.g. confidence intervals) |
| ☐ | ☒ | For null hypothesis testing, the test statistic (e.g. *F*, *t*, *r*) with confidence intervals, effect sizes, degrees of freedom and *P* value noted<br>*Give P values as exact values whenever suitable.* |
| ☒ | ☐ | For Bayesian analysis, information on the choice of priors and Markov chain Monte Carlo settings |
| ☒ | ☐ | For hierarchical and complex designs, identification of the appropriate level for tests and full reporting of outcomes |
| ☒ | ☐ | Estimates of effect sizes (e.g. Cohen's *d*, Pearson's *r*), indicating how they were calculated |

*Our web collection on statistics for biologists contains articles on many of the points above.*

## Software and code

Policy information about availability of computer code

| Data collection | bwa-v0.6.1, OxCal version 4.4.2, contamMix version 1.0–12, IntCal20, SHCal20, Yfull version 8.09, HaploGrep2, https://github.com/DReichLab/ADNA-Tools and https://github.com/DReichLab/adna-workflow |
|---|---|
| Data analysis | ADMIXTOOLS version 7.0.2, Eigensoft version 7.2.0, ADMIXTURE version 1.23, DATES |

For manuscripts utilizing custom algorithms or software that are central to the research but not yet described in published literature, software must be made available to editors and reviewers. We strongly encourage code deposition in a community repository (e.g. GitHub). See the Nature Portfolio guidelines for submitting code & software for further information.

## Data

Policy information about availability of data

All manuscripts must include a data availability statement. This statement should provide the following information, where applicable:

- Accession codes, unique identifiers, or web links for publicly available datasets
- A description of any restrictions on data availability
- For clinical datasets or third party data, please ensure that the statement adheres to our policy

SNP array genotype data for 122 modern individuals newly reported in this study can be obtained from the Harvard Dataverse repository through the following link (https://doi.org/10.7910/DVN/NC28XW). BAM files of aligned reads can be obtained from the European Nucleotide Archive (accession no. PRJEB58698).

## Human research participants

Policy information about studies involving human research participants and Sex and Gender in Research.

| | |
|---|---|
| Reporting on sex and gender | The Madagascar samples were obtained from only male participants to allow joint analysis of nuclear genome variation and Y chromosome variation. The Emirati samples were obtained from any volunteers and gender or sex were not a factor in sample collection. |
| Population characteristics | For newly reported Madagascar genotype samples, individuals were chosen who would likely have a long-standing history in the region, namely older males with a maternal grandmother and a paternal grandfather also from the same area.<br><br>Emirati samples were collected from healthy adults as controls for a study on rare disease. |
| Recruitment | Individuals who participated in sample collection provided informed consent. and sampling strategies are as described above. |
| Ethics oversight | The Madagascar sampling was approved by Human Subjects' Ethics Committees of the Health Ministry of Madagascar and by French committees (Ministry of Research, National Commission for Data Protection and Liberties and Persons Protection Committee).<br><br>The Emirati sampling was approved by the Al Ain District Human Research Ethical Committee. |

Note that full information on the approval of the study protocol must also be provided in the manuscript.

# Field-specific reporting

Please select the one below that is the best fit for your research. If you are not sure, read the appropriate sections before making your selection.

☒ Life sciences          ☐ Behavioural & social sciences          ☐ Ecological, evolutionary & environmental sciences

For a reference copy of the document with all sections, see nature.com/documents/nr-reporting-summary-flat.pdf

# Life sciences study design

All studies must disclose on these points even when the disclosure is negative.

| | |
|---|---|
| Sample size | We did not predetermine sample size. Our analyses are based on the data we were able to successfully collect from ancient individuals, which is always difficult. Standard errors are reported to describe uncertainty ranges of our analyses that often reflects both sample size and data quality per sample. |
| Data exclusions | We exclude data with low coverage and data that comes from individuals that are first (or second) degree relatives of other individuals. |
| Replication | We observe similar ancestry patters among different ancient individuals from the same and different areas. We are also able to replicate sequence data from ancient individuals for which we have multiple libraries processed. |
| Randomization | We are unable to randomize historical events, and so we expect correlation between individuals. However, our formal modeling methods use a block jackknife to obtain a meaningful estimate of the uncertainty of our inference through analysis of uncorrelated segments of the genome each of whcih provides an independent view into past history. |
| Blinding | We contextualized our individuals within an archaeological framework in order to set up our analyses (such as setting up groups based on location and probable proxy source populations). Blinding was not applicable in this study since the archaeological context is important. |

# Reporting for specific materials, systems and methods

We require information from authors about some types of materials, experimental systems and methods used in many studies. Here, indicate whether each material, system or method listed is relevant to your study. If you are not sure if a list item applies to your research, read the appropriate section before selecting a response.

## Materials & experimental systems

| n/a | Involved in the study |
|---|---|
| ☒ | ☐ Antibodies |
| ☒ | ☐ Eukaryotic cell lines |
| ☐ | ☒ Palaeontology and archaeology |
| ☒ | ☐ Animals and other organisms |
| ☒ | ☐ Clinical data |
| ☒ | ☐ Dual use research of concern |

## Methods

| n/a | Involved in the study |
|---|---|
| ☒ | ☐ ChIP-seq |
| ☒ | ☐ Flow cytometry |
| ☒ | ☐ MRI-based neuroimaging |

# Palaeontology and Archaeology

**Specimen provenance**

Permits for collection and sequencing of each individual or group of individuals was obtained and described in the supplemental information as:

"Songo Mnara: Samples were selected from excavated skeletal remains at Songo Mnara. Samples from adult burials included 1-2 teeth for isotopic study as well as one metatarsal or metacarpal for aDNA analysis. Infants were not sampled during excavations. One infant mandible recovered during analysis of domestic faunal remains was also subjected to aDNA extraction. Teeth samples are stored at the University of Bristol. Bone samples are currently with the Reich laboratory at Harvard University; after this study is complete, they will be returned to Tanzania via the National Museums of Tanzania.
Kilwa and Lindi: For Kilwa and Lindi, samples were selected following protocols outlined by Prendergast and Sawchuk (2018) and Prendergast et al. (2019).

For Kilwa, three individuals were identified by E. Sawchuk among the commingled and largely unlabeled remains. Two samples were collected for one individual (KS.01.01, KS.01.02; the latter was not sampled and was returned intact), and one for each of the other individuals (KS.02.01, KS.03.01). Samples from Kilwa were exported and returned to the NMK by co-author C. Ogola in October 2017, and subsequently returned to the British Institute in East Africa (BIEA) where the rest of the remains are curated by co-author Dr. E. Ndiema in 2022. Arrangements have been made to repatriate all Kilwa human remains from the BIEA to the National Museum of Tanzania in Dar es Salaam.

For Lindi, two samples were chosen from the single individual present at this site, an adult male: the petrous portion of the left temporal bone (sample number LS1.02); and the right upper central incisor (LS1.01). Only the petrous was sampled, and the remainder of this petrous, and the complete unsampled incisor, were both returned to the NMT by M. Prendergast, along with a complete record of sampling activity and aDNA and C14 results, in May 2019. As described in a Memorandum of Agreement (MOA) between the Reich lab and the National Museum of Tanzania, the remaining powder, DNA extracts, and libraries remain under curation at the Reich Laboratory at Harvard University.

Mtwapa
Bone and tooth samples were taken over the course of the excavations at the site, which began under the direction of Kusimba in 1996. Most of the samples were collected by Monge and Williams during the 2010 season, however, when excavations focused on the cemetery located near the mosque. Tooth samples were collected for radiocarbon dating, stable isotope, and genetic analyses. Long bone fragments and rib samples were collected for stable isotope analyses. After each excavation season, the human remains were reburied, except for one individual remains are curated at the NMK. Any sample material remaining after analysis either has been or will be returned to Kusimba for curation at the NMK.

Manda
All samples were collected during a single excavation season in 2011-2012. Tooth samples were collected by co-authors J. Monge and S. Williams for radiocarbon dating, stable isotope, and genetic analyses. Long bone fragments and rib samples were collected for stable isotope analyses. The human remains were reburied the following year when osteological analyses were completed. Any sample material remaining after analysis either has been or will be returned to co-author C.M. Kusimba for return to the NMK.

Makwasinyi
Tooth fragment samples were collected by co-author C.M. Kusimba. The crania were left undisturbed in their respective localities. Any remaining sample material has been returned to co-author C.M. Kusimba for return to the NMK."

**Specimen deposition**

See above.

**Dating methods**

We provide new dates for several ancient individuals to understand the context of the admixture patters we detect and to help us build a chronology of these individuals. We obtained dating results from the Pennsylvania State University Radiocarbon laboratory or the Illinois State Geological Survey. Radiocarbon ages were calibrated "in OxCal version 4.4.2 (Bronk Ramsey (2009), using either the IntCal20 (Reimer et al 2020) or SHCal20 (Hogg et al. 2020) calibration latitude depending on whether the site was north (Kenya) or south (Tanzania) of the equator."

☒ Tick this box to confirm that the raw and calibrated dates are available in the paper or in Supplementary Information.

**Ethics oversight**

We obtained all permissions necessary for ancient DNA analysis from the respective countries from where the ancient individuals were buried or kept (see above), and we worked directly with communities that lived near the archaeological site and have traditions of inheritance from the ancient communities.

Note that full information on the approval of the study protocol must also be provided in the manuscript.

