## [Peer Review File · Nature]

Manuscript Title: Entwined African and Asian Genetic Roots of Medieval Peoples of the Swahili Coast

Reviewer Comments & Author Rebuttals

Reviewer Reports on the Initial Version:

Referees' comments:

Referee #1 (Remarks to the Author):

The manuscript from Brielle et al., investigates the genetic ancestry of Medieval and early modern coastal Swahili individuals from wealthy and elite context in Kenya and Tanzania as well as individuals from an early modern site in inland Kenya. The authors compared the results of their historical samples to newly generated data from present-day individuals from the United Arab Emirate and Madagascar as well as previously published data of present-day Swahili people from Kenya. The study shows that the trade across eastern Africa and the Indian ocean, which started already in the first millennium CE, was accompanied by genetic admixture between Eastern Africans and Asians. All Swahili individuals from medieval and early modern sites in Kenya have mixed African and Persian related (Persian-Indian or Persian-Arabian) ancestry. Medieval individuals from Tanzania also show similar instances of admixture but display a more cosmopolitan pattern. Interestingly, 76% to 100% of the African ancestry of the admixed individuals derived from female ancestor while 100% of the Asian ancestry derived from male ancestors. The authors dated the admixture event between the African females and the Asian males between ~938 and ~1242 CE. They also show that present-day coastal Swahili people in Kenya have larger proportion of African ancestry in than their medieval counterparts and are more similar to historical individuals from inland Kenya. However, wealthier present-day Swahili groups seems to have preserved to a lesser extend the ancestry pattern observed in historical Swahili individuals from elite context.

The study is interesting as it provides the first insights of the genetic consequences of the trade in Coastal East Africa. The manuscript gives the feeling that the authors hoped to obtain a more complete narrative of coastal identity through time from the data. However, the dataset is biased as 98.5% of the individuals seem to belong to elite group and are not representative of the overall medieval coastal Swahili population. They try to push the narrative of autochthonous development of coastal settlement in opposition to the narrative constructed by mid-20th century colonialist archaeologists, but that narrative is not clearly mirrored in the data. I hope the following comments will help to bring more focus on the new set of insights revealed by the data and highlight even more the interesting story they are bringing to our knowledge.

General major comments

1. The first paragraph of the introduction is not enough for the reader to understand what is refer to as "Swahili people". I'm afraid many don't know if it is an ethnical group or not, their geographical distribution nowadays as well as their political and social status. A good understanding of what the Swahili culture represents will help to build the momentum and to stir up the reader's interest.
2. The introduction should be re-worked, it is going back and forth in time and seem to bring down

different ideas into multiple questions. However, the questions have all the same scope and question the genetic origin of ancient coastal Swahili individuals. I think it is redundant and confusing at part. The introduction will be more stimulating if different paragraphs were built in order the guide the reader toward specific inferences made from the dataset.

3. I urge the authors to refer to different populations and group of population in a more consistent way in the text and the figures. Sometimes a population name is reported in the figure but is referenced in the text by the group they represent. Figure 1b reports groups, population name and country but Figure 1c reports only population names. This makes it difficult to link the different analyses together.

4. I would like more clarifications about the constraints of using different references populations/individual to estimate the same ancestry in different individuals. In the study the African ancestry is better modelled using the Makwasinyi as reference for the Kenyan individuals and the Lindi as reference for the Tanzanian individuals. Are the proportion of African ancestry estimated using different references comparable?

5. As the authors stated, a limitation of this type of study is the choice of reference population which doesn't, in most of the case, represent the precise source of ancestry. How the divergence between the proxy population and the precise population source may affect the ancestry proportion estimated? I urge the author to be more cautious in the interpretation of the proportion of African and Asian ancestry in the admixed individuals. In my understanding, an unequal distance between the references used and the precise source of ancestry may under or overestimate the proportions. In my opinion the proportion of African and Asian ancestry in ancient Kenyan individuals is closer to 50-50 than what the authors stated. By emphasizing that "the people of African ancestry made the largest genetic contribution to the mixed population" I'm wondering if the authors are not biased toward bringing more arguments to support the narrative of autochthonous origin of the coastal Swahili lifestyle.

Specific major comments

1. The Admixture analyses show a low Asian-like ancestry in Makwasinyi individuals in comparison to the Lindi individual which are exclusively African-related ancestry. The qpADM modelling fit best the Makwasinyi as 100% African. I would like the author to address these observations. Furthermore, the ternary plot shows that the Makwasinyi has "nearly" 0% of Indian-persion ancestry... what "nearly" means here?

2. The authors are very cautious about interpreting the African-Asian admixture of medieval and early modern Swahili coastal individuals as deriving from a single event over few generations or one drawn out over dozens of generations. However, the historic evidence of presence of Persians and Arabs settlers on the Eastern African coast over several centuries, the estimation of date of admixture with confidence intervals overlapping 3 to 8 centuries, and the change of the Asian ancestry in mixed individuals from a Persian-Indian during medieval time to a Persian-Arab ancestry during early modern time suggest to me several events of admixture drawn over dozens of generations.

General minor comments

1. The most striking difference between medieval and present-day Swahili people of the Eastern African coast is the higher proportion of bantu-related ancestry in present-day people. As the author observed that the genetic profile of present-day coastal individual is more similar to early modern

inland group, I would like the author to discuss/investigate deeper whether the change of Bantu-related ancestry is an effect of the biased dataset of medieval coastal individuals, a replacement of people of medieval people at the coast by inland groups or further admixture events between coastal and inland which diluted the Asian ancestry of people in the coast after the 16th century.

2. The results show population structure in coastal medieval and early modern Swahili individuals. Are there some social models that can be proposed to fit the observation of the elite medieval and early modern coastal group maintaining an almost 50-50 proportion of African and Asian ancestry and a male-female participation pattern over generations?

Specific minor comments

1. Abstract stated that genome wide data of 80 individuals are reported, while it is genome wide data of 54 individuals out of 80 analyzed.
2. In the abstract, the sentence "This Asian component was approximately eighty to ninety percent from Near Eastern males and ten to twenty percent from Indian females" (L55-56) is confusing as it is not clear whether the proportion provided applied to each ancestry components, to the sex of the individuals that contributes the each ancestry or to both. I suggest to the author to rewrite the sentence for clarification.
3. Is figure 1A represents the geographic distribution of present-day or Medieval/early modern period or both?
4. The font size on most figures is way too small, almost impossible to read on printed A4 format.
5. Sentence L79 to L81, the authors state that research demonstrated the "autochthonous development of coastal settlement". That is appearing as a too strong statement as some interrogations remain hence the need to investigate the genetic origins of past people of that region.
6. Sentence L98 to L100, about movement of people from central region of Africa to the coast mentions movement of people from "the settlement of people from the Yemenite region of Hadramawt"? I'm not sure I got the relevance or that I understand the Yemenite mention right. Can you rewrite the sentence for clarity.
7. Sentence L115 to L116, I suggest the author to incorporate in the text one or two evidence from reference 10 which have been ignored by colonialist archaeologists and support the complex society created by African people. Maybe the paragraph 109 to 117 and 119-128 can be merged.
8. The tridimensional PCA of figure 1B, is a great idea in theory but in practice the figure is very difficult to read. I'm not sure the eigenvector 3 bring much more information. The 2 dimensional PCA of extended figure 3 is much more easy to grasp and the outcome cannot be clearer. I suggest to the author to replace the tridimensional PCA of figure 1B by the one in extended figure 3A.
9. Extended table 1 list the individual, their sex and mitochondrial and Y chromosome haplotype, however the legend of the table detailed the family ties of related individual. The legend doesn't correspond to the table. As it might be of interest for some readers, I suggest the author to add an extended figure or table portraying the family ties between the new ancient individuals.
10. Extended data table 3 – line and column are not always aligned.
11. The Lindi individual cannot be seen in any of the PCA as he probably cluster with the bantu and is represented by the same color. Choose another brighter color for the Lindi individual.
12. It is reported that 4 individuals for Manda passed the non first degree relative filter, but only 3 mtDNA are included in Table 1. Is the fourth individual I17939 for which only less than 2x coverage of the MT genome was recovered? If yes, it should be you mentioned in the legend of table 1.
13. The sentence L351 which state that males Indian-Persian were on the coast already by ~1000 CE

and mixed “further” with female sub-saharan Africans females, gives the feeling that the admixture happened after 1000 CE. The lower interval of the date of admixture is 938 CE when considering all 3 groups and can go down to 614 CE. I suggest to the author to rephrase the sentence to not confuse the reader.

Referee #2 (Remarks to the Author):

Brielle et al present a study of 80 new ancient DNA samples from multiple coastal sites in Kenya and Tanzania. The main question of their study centered around if there is evidence of non-African admixture in ancient samples from the Swahili Coast. The authors applied multiple population genetics methodologies such as ADMIXTURE and qpAdm to model the ancestry/admixture history of the newly reported ancient DNA samples. The authors found evidence of admixture in many of the coastal samples related to a combined source of Persian and Indian ancestry. They used a method called DATES to estimate that this admixture had started by at least 1000 CE. The authors also examined sex-biased admixture dynamics by analyzing mitochondria/Y chromosome haplogroups, and comparing ancestry estimates between chr1-22 and chrX. This suggested that most of the Persian ancestry was from males and most of the African ancestry was from females. The authors methodology to answering these questions is appropriate and their conclusions are well supported. Their dataset is highly novel and the questions they address give this manuscript a high impact on the fields of population genetics and anthropology. However, prior to publication there are multiple comments that the authors need to address:

1. The qpAdm modeling is very thorough in examining multiple combinations of admixing populations. However, I think that there are multiple areas that should be clarified to help a general audience interpret these results:

a. The supplemental section describing the qpAdm modeling presents many different combinations of populations that were used. However, I found it a little challenging in some cases to determine how some models were rejected or why certain models were broken. It might be helpful to include additional descriptions about the criteria that is used to consider a model broken/rejected or what makes a model a better fit than an alternative working model. For example on lines 600-601,

"whereas the Indian population breaks a model with the East Asian populations" Is this due to $P = 0.000000$ in all entries for the column "P value for East Asian population on left, Ulladan on right?"

b. Lines 216-217: “Any model that does not include all three ancestries can be rejected with high statistical confidence.” It is not clear from the main text which table/figure includes this data and what is considered high statistical confidence.

c. What is the significance of the three lines that are shaded in grey in Extended Data Table 2? Also the same for table S7.

d. What is the "POP" column in Table S7?

2. How was $K = 8$ chosen as the ADMIXTURE run to present in the main text? Was cross validation considered or were other criteria applied for selecting this value of K ? What criteria did you use for selecting one of the four replicates to represent each value of K ?

3. How was the relatedness filtering performed on the new ancient dna samples? Could you include

a short description in the methods section describing how relatedness was determined?

4. Main text lines 153-154: "We also radiocarbon dated enough individuals from each site to establish chronology" Can you clarify how many samples are meant by "enough?"

5. Main text lines 330-332: "Under the simplifying assumption that the mixture occurred in a single event over just a few generations, rather than over a period that spanned many generations" How would your results be impacted if this assumption is incorrect?

6. On line 379 in the main text: "After removing two outliers (according to the ADMIXTURE graph." I think this is described in the supplement, but it is not clear from the main text what an outlier sample is from your ADMIXTURE analysis and how that was determined.

7. Main text line 882: "haplogroups were determined according to the Yfull tree" Is there a citation for "Yfull tree?"

8. Bellow are a few grammar/spelling comments to clarify:

a. Supplement line 652: "ancestry, degraded the model generally by one two orders of magnitude." Should this be "one or two?"

b. Supplement line 1144: "Kilwa individual to use as a reference, and so we used archaeological context dte range of 1300." I think "dte" should be "date."

Referee #3 (Remarks to the Author):

A. The paper outlines the outcomes of the genetic analysis of 80 individuals from six sites across Kenya and Tanzania - of which 54 were included in the analysis. These individuals are associated with dates 1300-1800CE. These are compared to 13 samples from Makwasinyi, from 1650-1950CE considered as a proxy for contemporary inland communities. The data suggest some differences between the northern Swahili coast and Southern Swahili coast - with those individuals from northern sites showing African/Persian/Indian ancestry; while individuals from southern sites showing Arabic ancestry - particularly those from later periods. Individuals from Kilwa Songo Mnara and Lindi return strong Bantu signature, while those from northern sites the African component appears to be a mixture of Bantu and Pastoralist-related ancestry - consistent with the Makwasinyi samples.

Analysis suggests that foreign DNA was likely introduced through the males, with male ancestors mostly from Persia, and female ancestors mostly African or Indian.

B. The study demonstrates a novel approach to examining the question of Swahili origins and represents a significant new body of work.

C. The study presents a reasonable cumulative dataset, however it is heavily skewed to Mtwapa over the other sites assessed. The scale of the assemblages is also slightly confusing. The paper introduces

a sample of 80 individuals. They then remove 26 for various reasons (poor samples, relatives to others etc). It seems 67 individuals provided radio carbon dates. They note dated samples represent 24 (44%) of the 54 remaining individuals.

In figure 1 51 dated samples are shown 10 from Makwasinyi and the remaining 41 from the other assemblages. In figure 2 the 10 from Makwasinyi were excluded. However, I am not clear what happened to the other 13 samples from individuals of sufficient quality for assessment. I initially thought this was because the individuals presented in the figures had dates associated but there seem to be too many then. This is confusing.

I do think more transparency is needed as to why the majority of the individuals selected post date 1300. Analysis of earlier populations would have allowed the authors to explore whether or how genetic ancestry changed once there was greater connection with the Indian Ocean World.

The methods all seem appropriate, but do assume a lot of prior knowledge of genetic analysis - this may be appropriate for the readership, but it is quite jargon heavy. The data analysis is quite dense which can make it very difficult to read. This is particularly apparent in the section on genetic ancestry by Sex paragraphs lines 284-236. Thinning out and making the methods more accessible to a wider readership seems important as this paper is likely to have appeal not just to those working on Genetics, but those looking at the archaeology the East African coast more broadly.

I think clarity may be slightly impacted by the decision to present the ancestral groupings overall rather than the data from each of the sites individually first. I think this would have helped tease out both the chronological and regional variation more clearly. In the current form you jump around from site to site - which can be hard to follow - especially if you didn't already have a clear knowledge of where the sites are.

Line 270/271 - I think you've mixed up the dark yellow and light yellow in your caption?

In the data section the authors reference two previous studies Raaum et al [29] and Brucato et al [28]. I think the work could benefit from a summary of these studies as it is not clear how comparable they are.

D. Statistics and treatment of uncertainties are clear, though I do think the explanation for some of the outliers as indicative of the cosmopolitan nature of the region is a bit over general.

E. The conclusions draw on a wider range of sources (historical, linguistic, etc) to interpret the datasets presented. The limitations of the study - in terms of the selection of burials from elite contexts and associated with Islamic burials are clear.

F. None additional to the comments above.

G. References are appropriate.

H. The abstract is clear and appropriate, though it does suggest a larger sample than is actually included in the analysis. The introduction and conclusions are clear.

Author Rebuttals to Initial Comments:

Referee #1

The manuscript from Brielle et al., investigates the genetic ancestry of Medieval and early modern coastal Swahili individuals from wealthy and elite context in Kenya and Tanzania as well as individuals from an early modern site in inland Kenya. The authors compared the results of their historical samples to newly generated data from present-day individuals from the United Arab Emirate and Madagascar as well as previously published data of present-day Swahili people from Kenya. The study shows that the trade across eastern Africa and the Indian ocean, which started already in the first millennium CE, was accompanied by genetic admixture between Eastern Africans and Asians. All Swahili individuals from medieval and early modern sites in Kenya have mixed African and Persian related (Persian-Indian or Persian-Arabian) ancestry. Medieval individuals from Tanzania also show similar instances of admixture but display a more cosmopolitan pattern. Interestingly, 76% to 100% of the African ancestry of the admixed individuals derived from female ancestor while 100% of the Asian ancestry derived from male ancestors. The authors dated the admixture event between the African females and the Asian males between ~938 and ~1242 CE. They also show that present-day coastal Swahili people in Kenya have larger proportion of African ancestry in than their medieval counterparts and are more similar to historical individuals from inland Kenya. However, wealthier present-day Swahili groups seems to have preserved to a lesser extend the ancestry pattern observed in historical Swahili individuals from elite context.

The study is interesting as it provides the first insights of the genetic consequences of the trade in Coastal East Africa. The manuscript gives the feeling that the authors hoped to obtain a more complete narrative of coastal identity through time from the data. However, the dataset is biased as 98.5% of the individuals seem to belong to elite group and are not representative of the overall medieval coastal Swahili population. They try to push the narrative of autochthonous development of coastal settlement in opposition to the narrative constructed by mid-20th century colonialist archaeologists, but that narrative is not clearly mirrored in the data. I hope the following comments will help to bring more focus on the new set of insights revealed by the data and highlight even more the interesting story they are bringing to our knowledge.

General major comments

1. The first paragraph of the introduction is not enough for the reader to understand what is refer to as "Swahili people". I'm afraid many don't know if it is an ethnical group or not, their geographical distribution nowadays as well as their political and social status. A good understanding of what the Swahili culture represents will help to build the momentum and to stir up the reader's interest.

Present-day "Swahili" identity and how it relates to the medieval and early modern Swahili culture, is uncertain. Following the reviewer's recommendation, we have rewritten the first paragraph to explain this conundrum better and to provide a clearer motivation for some of our research:

"The Swahili culture of eastern Africa is defined by a set of shared features: a common language of African origin (Kiswahili), a shared predominant religion (Islam), and a geographic distribution in coastal towns and villages. People of the medieval and early modern Swahili culture lived over a vast region that included northern Mozambique to present-day southern Somalia, Madagascar, and the archipelagoes of Comoros, Kilwa, Mafia, Zanzibar, and Lamu (yellow outlines in **Error! Reference source not found.**A) [1, 2]. Millions of coastal people today identify as Swahili, although for many this is a secondary identity with primary identities often being more based on town of origin, family history, or traditional social status [3]. How people who identify as Swahili today relate to people of the medieval and early modern Swahili culture has been difficult to elucidate in the absence of ancient DNA."

2. The introduction should be re-worked, it is going back and forth in time and seem to bring down different ideas into multiple questions. However, the questions have all the same scope and question the genetic origin of ancient coastal Swahili individuals. I think it is redundant and confusing at part.

The introduction will be more stimulating if different paragraphs were built in order to guide the reader toward specific inferences made from the dataset.

We have heavily reworked the introduction for clarity, with the goal of more clearly guiding the readership toward understanding the research questions that we address through genetic analysis.

3. I urge the authors to refer to different populations and group of population in a more consistent way in the text and the figures. Sometimes a population name is reported in the figure but is referenced in the text by the group they represent. Figure 1b reports groups, population name and country but Figure 1c reports only population names. This makes it difficult to link the different analyses together.

We have fixed these inconsistencies in the figures.

4. I would like more clarifications about the constraints of using different reference populations/individual to estimate the same ancestry in different individuals. In the study the African ancestry is better modelled using the Makwasinyi as reference for the Kenyan individuals and the Lindi as reference for the Tanzanian individuals. Are the proportion of African ancestry estimated using different references comparable?

Our modelling reveals a statistically significant difference in the type and proportion of sub-Saharan African-related ancestry between the people of the northern towns and those of the southern towns. Thus, while we are able to quantify the proportion of African ancestry in the group of individuals in each coastal town—and are very clear about those proportions in our manuscript—the sources of that ancestry are different, and it is important to be clear about that as well. The most appropriate proxy sub-Saharan African ancestry source for both the northern Kenyan coastal groups and the southern Tanzanian coastal groups is Bantu-associated, and Pastoral Neolithic-associated ancestry is also found at a significant proportion in the northern coastal groups. The Mtwapa and Manda groups could not be successfully modelled with a single Bantu-associated or a single Pastoral Neolithic-associated group. However, they could be successfully modelled with the Makwasinyi group, which itself carried ancestry from the above two ancestry types. The Makwasinyi group, however, was not a fitting proxy for the ancestry of the Tanzanian individuals who our analysis suggests have little or no Pastoral Neolithic-associated ancestry.

5. As the authors stated, a limitation of this type of study is the choice of reference population which doesn't, in most of the case, represent the precise source of ancestry. How the divergence between the proxy population and the precise population source may affect the ancestry proportion estimated?

I urge the author to be more cautious in the interpretation of the proportion of African and Asian ancestry in the admixed individuals. In my understanding, an unequal distance between the references used and the precise source of ancestry may under or overestimate the proportions. In my opinion the proportion of African and Asian ancestry in ancient Kenyan individuals is closer to 50-50 than what the authors stated. By emphasizing that "the people of African ancestry made the largest genetic contribution to the mixed population" I'm wondering if the authors are not biased toward bringing more arguments to support the narrative of autochthonous origin of the coastal Swahili lifestyle.

The reviewer raises the question of what kinds of biases might arise from the use of proxy source populations in genetic analysis. Systematic uncertainties associated with using proxy sources (instead of the "actual" source which it is never possible to sample) will increase error in estimates. However, the key motivation for the *qpAdm* methodology which we use for the estimates is to enable estimates of ancestry proportions that are unbiased by the fact that proxy groups differ in their sample sizes and distance from the actual sources. That pitfall the reviewer highlights is known to bias methodologies like ADMIXTURE (for a detailed discussion of this issue, see "A tutorial on how not to over-interpret STRUCTURE and ADMIXTURE bar plots", Lawson, vanDorp, and Falush, Nature Communications 2018), and *qpAdm* has been proven to solve these issues. In more detail, the *qpAdm* methodology leverages genetic drift deep in the past before the proxy and actual source diverged. Because of this, *qpAdm* is not expected to be biased by differences that arose between proxy populations and true source populations since their divergence. Estimating the proportions of shared drift with the different proxies is therefore expected to produce unbiased estimates, while limited sample size, or divergence

between the source and proxy, will not bias estimates even while increasing statistical uncertainty. The key point is that we can precisely determine the uncertainty (standard errors), and find it is much smaller than the difference in African- and Asian-associated ancestry proportions.

We agree that a substantial number of medieval individuals we analyzed have >50% Near Eastern ancestry. In our revision, we no longer state that “people of African ancestry made the largest genetic contribution to the mixed population”, and instead write “People of both African and Asian ancestry made major contributions to the mixed population, with African proportions being ~57% on average at Mtwapa and Faza, ~32% at Manda, ~67% at Songo Mnara, and ~74% at Kilwa”.

Specific major comments

1. The Admixture analyses show a low Asian-like ancestry in Makwasinyi individuals in comparison to the Lindi individual which are exclusively African-related ancestry. The qpADM modelling fit best the Makwasinyi as 100% African. I would like the author to address these observations. Furthermore, the ternary plot shows that the Makwasinyi has “nearly” 0% of Indian-persian ancestry... what “nearly” means here?

We agree that this issue is confusing and have clarified it in our revision through an added paragraph. Specifically, we now specify that when we refer to “African” ancestry, we mean ancestry from people who likely lived in sub-Saharan Africa 4000-1000 years ago (as documented directly in this case with ancient DNA from that period). The fact that the ancestors of some of these African source populations 4000-1000 years ago may have come from West Asia deeper in time is not a contradiction to our definition, as we are talking about proximate source populations at a specific time depth. All humans are mixed at multiple time depths of human history, so to define the proportion of genetic ancestry, one needs to be specific about time as well as geography (see “What is ancestry?” Mathieson and Scally, PLoS Genetics 2020, which we now reference).

The confusing issue the reviewer refers to arises because ADMIXTURE is detecting components of ancestry in Makwasinyi that are closely related to ones present at high frequency in West Asia. However, our modelling shows that what is represented in this way in ADMIXTURE in Makwasinyi is in fact likely statistically consistent with being derived from Pastoral Neolithic (PN)-associated populations, which were widespread in East Africa as documented by ancient DNA by 4000 years ago. Previous work has shown that these PN populations have ~40% ancestry potentially derived from groups related to West Asians at a time depth of >4000 years ago (Science. 2019 Jul 5; 365(6448): eaaw6275). We have updated the ADMIXTURE plot to include such Pastoral Neolithic populations for reference (to show that thousands of years ago sub-Saharan Africans also had this ancestry profile) and to explicitly clarify why the plot is not implying Asian ancestry in Makwasinyi.

2. The authors are very cautious about interpreting the African-Asian admixture of medieval and early modern Swahili coastal individuals as deriving from a single event over few generations or one drawn out over dozens of generations. However, the historic evidence of presence of Persians and Arabs settlers on the Eastern African coast over several centuries, the estimation of date of admixture with confidence intervals overlapping 3 to 8 centuries, and the change of the Asian ancestry in mixed individuals from a Persian-Indian during medieval time to a Persian-Arab ancestry during early modern time suggest to me several events of admixture drawn over dozens of generations.

We agree this aspect of our originally submitted manuscript was confusing. In our revision, we make it crystal-clear that the genetic data shows that the admixture did not occur all at once, and instead show that it was drawn out over dozens of generations, with direct evidence for this including: (a) an admixture date with an overlap of 795-1085 CE meaning that mixture with Asians must have begun by this time, (b) a change in the source of Asian ancestry from more Persian-related in the Manda site to more Arab-related in some Mtwapa individuals and some of the later Tanzanian individuals, and (c) significant heterogeneity in ancestry within sites which means that the Swahili coastal populations were not in Hardy-Weinberg equilibrium even in the late medieval period and even in the same towns and were continually incorporating new migrants of entirely African or West Asian admixture. In our revision, we clarify that when we refer to the events ~1000 years ago, we are focusing on the initial period of admixture, and cannot distinguish between a scenario of a short period of initial admixture responsible for most of the West Asian ancestry in the region in medieval times (before 1500 CE), and

one that was drawn out over a longer period. An important topic for future ancient DNA work, outside of the scope of this study, is to distinguish these scenarios.

General minor comments

1. The most striking difference between medieval and present-day Swahili people of the Eastern African coast is the higher proportion of bantu-related ancestry in present-day people. As the author observed that the genetic profile of present-day coastal individual is more similar to early modern inland group, I would like the author to discuss/investigate deeper whether the change of Bantu-related ancestry is an effect of the biased dataset of medieval coastal individuals, a replacement of people of medieval people at the coast by inland groups or further admixture events between coastal and inland which diluted the Asian ancestry of people in the coast after the 16th century.

We have robustly addressed these issues in our revision.

Specifically, we now report new whole-genome genotyping data from 93 present-day individuals who reported long family residency in coastal towns (specifically, the sampling was from the modern coastal towns of Faza, Kizingitini, Ndau, Pate, Wasini, and Jomvu Kuu). In our original manuscript, we reported mitochondrial and Y chromosome haplogroup frequencies from the set of individuals sampled by Raaum et al. (using published tabulations from "Decoding the genetic ancestry of the Swahili" Raaum et al. Routledge. p. 81-102). Our original manuscript highlighted differences in the haplogroup frequencies between the Raaum et al. sampling and the other sampling from which we had haplogroup frequency data as well as whole genome data ("The Comoros Show the Earliest Austronesian Gene Flow into the Swahili Corridor", Brucato et al. AJHG 2018), based on individuals who self-identified as Swahili speakers and who reported that their grandparents were ethnically Swahili, but who did not necessarily report connections to peoples of the medieval coastal towns.

Our new whole genome data confirms large qualitative differences in the ancestry profiles between these two samplings, with much more Southwest Asian ancestry in the Raaum et al. data. We observe that the processes that shaped the genomes of the present-day individuals from the Raaum et al. sampling are qualitatively similar in many ways to those that shaped the genomes of the medieval individuals, while also reflecting additional events in the last centuries that had not yet impacted the medieval groups. This indicates that there was not large-scale replacement of the medieval population in the ancestry of some present-day Swahili-identified communities. It is also possible that we are observing class-based differences in both the present-day and medieval age. We have noted all of this in the revised manuscript.

Our newly added data make it even clearer than in our original manuscript that present-day people who identify as Swahili are genetically heterogeneous, with some having relatively more genetic continuity with the sampled individuals of the medieval coastal towns than others. There was continuity, dilution, and additional admixture from non-African and African sources since medieval times. The addition of these new data further strengthens our study and provides direct evidence for the complexity of the modern relationship (and in some cases disconnect) between modern Swahili identity and genetics.

2. The results show population structure in coastal medieval and early modern Swahili individuals. Are there some social models that can be proposed to fit the observation of the elite medieval and early modern coastal group maintaining an almost 50-50 proportion of African and Asian ancestry and a male-female participation pattern over generations?

The proportions at Mtwapa and Manda are not 50-50%, but instead variable across individuals and sites. The proportions at Lindi and Songo Mnara and Kilwa are also substantially different and variable. The high variation in proportions suggests to us that we do not have adequate sampling to argue that the proportions of mixture anywhere are close to 50-50%, so we do not think we have genetic data that are strong enough to motivate a discussion of the reasons for the proportions.

That said, we think this topic is interesting, and one speculative possibility (too speculative to discuss in the paper) is that a close to 50-50% mixture proportion might reflect a scenario in which the population was created by first-generation mixtures of Southwest Asian males and local African

females (which would be consistent with the extreme sex-biased mixture signals we detect), who then mixed with each other with only modest additional input mostly on the African side which could explain why the African proportions are on average higher. We would like to keep this genetics study focused on the robust genetic results which are very confident and think that such social modelling analyses could be of interest for future studies.

Specific minor comments

1. Abstract stated that genome wide data of 80 individuals are reported, while it is genome wide data of 54 individuals out of 80 analyzed.

The reasons for not including all 80 individuals in the whole genome analyses are that (a) some are genetically determined to be first degree relatives of another individual included in our main analysis and so not statistically very independent or (b) some had sequencing coverage that was relatively modest (<15,000 SNPs with data) making ultra-high-resolution analyses difficult, or (c) the data from them was less robust for use in our main analyses because of hints of contamination. However, all are scientifically valuable, providing additional information for example by providing confident mitochondrial DNA haplogroups, or allowing determination of family relationships, so we continue to refer to these individuals in the abstract. We edited the relevant two sentences to make it clear that we perform high-resolution analysis on only a subset of individuals:

“We report ancient DNA data from 80 individuals deriving from six medieval and early modern (1250-1800 CE) coastal towns as well as from an inland town postdating 1650 CE. Many coastal individuals for whom we could perform high-resolution analysis had over half of their DNA derived from primarily female African ancestors, with large proportions and occasionally more than half coming from Asian ancestors.”

2. In the abstract, the sentence “This Asian component was approximately eighty to ninety percent from Near Eastern males and ten to twenty percent from Indian females” (L55-56) is confusing as it is not clear whether the proportion provided applied to each ancestry components, to the sex of the individuals that contributes the each ancestry or to both. I suggest to the author to rewrite the sentence for clarification.

We have reworded this sentence, which now reads: “The Asian ancestry included both Persian-associated and Indian-associated components, with eighty to ninety percent deriving from Persian males.”

3. Is figure 1A represents the geographic distribution of present-day or Medieval/early modern period or both?

Figure 1A represents the geographic locations of ancient coastal individuals investigated in this study (not the present-day individuals). We have clarified this in the caption.

4. The font size on most figures is way too small, almost impossible to read on printed A4 format.

We have increased the font sizes in the figures.

5. Sentence L79 to L81, the authors state that research demonstrated the “autochthonous development of coastal settlement”. That is appearing as a too strong statement as some interrogations remain hence the need to investigate the genetic origins of past people of that region.

We no longer use the word “autochthonous,” and continue to emphasize the mixed African and Asian roots of the analyzed medieval and modern individuals.

6. Sentence L98 to L100, about movement of people from central region of Africa to the coast mentions movement of people from “the settlement of people from the Yemenite region of Hadramawt”? I’m not sure I got the relevance or that I understand the Yemenite mention right. Can you rewrite the sentence for clarity.

We have now added clarification to that sentence to state that there was settlement of people from the Yemenite region of Hadramawt *to the African coast*:

“In the 19th century, growth of overseas trade, including in enslaved people, led to large-scale population movements from central regions of Africa and settlers from the Yemeni region of Hadramawt to the Swahili coast.”

7. Sentence L115 to L116, I suggest the author to incorporate in the text one or two evidence from reference 10 which have been ignored by colonialist archaeologists and support the complex society created by African people. Maybe the paragraph 109 to 117 and 119-128 can be merged.

We have added in examples.

8. The tridimensional PCA of figure 1B, is a great idea in theory but in practice the figure is very difficult to read. I'm not sure the eigenvector 3 bring much more information. The 2 dimensional PCA of extended figure 3 is much more easy to grasp and the outcome cannot be clearer. I suggest to the author to replace the tridimensional PCA of figure 1B by the one in extended figure 3A.

We have replaced the 3-dimensional PCA in Figure 1B with a 2-dimensional PCA.

9. Extended table 1 list the individual, their sex and mitochondrial and Y chromosome haplotype, however the legend of the table detailed the family ties of related individual. The legend doesn't correspond to the table. As it might be of interest for some readers, I suggest the author to add an extended figure or table portraying the family ties between the new ancient individuals.

We have expanded the legend to make it clear which individuals from each family group were removed from the main analysis and we have also specified all individuals in each family in the table.

10. Extended data table 3 – line and column are not always aligned.

We have fixed this.

11. The Lindi individual cannot be seen in any of the PCA as he probably cluster with the bantu and is represented by the same color. Choose another brighter color for the Lindi individual.

We have changed the colors and shapes to make the PCA clearer.

12. It is reported that 4 individuals for Manda passed the non first degree relative filter, but only 3 mtDNA are included in Table 1. Is the fourth individual I17939 for which only less than 2x coverage of the MT genome was recovered? If yes, it should be you mentioned in the legend of table 1.

This is correct, and we now mention this in the legend of Table 1.

13. The sentence L351 which state that males Indian-Persian were on the coast already by ~1000 CE and mixed “further” with female sub-saharan Africans females, gives the feeling that the admixture happened after 1000 CE. The lower interval of the date of admixture is 938 CE when considering all 3 groups and can go down to 614 CE. I suggest to the author to rephrase the sentence to not confuse the reader.

We rephrased the sentence following the reviewer's suggestion.

Referee #2

Brielle et al present a study of 80 new ancient DNA samples from multiple coastal sites in Kenya and Tanzania. The main question of their study centered around if there is evidence of non-African admixture in ancient samples from the Swahili Coast. The authors applied multiple population genetics methodologies such as ADMIXTURE and qpAdm to model the ancestry/admixture history of the newly reported ancient DNA samples. The authors found evidence of admixture in many of the coastal samples related to a combined source of Persian and Indian ancestry. They used a method called DATES to estimate that this admixture had started by at least 1000 CE. The authors also examined sex-biased admixture dynamics by analyzing mitochondria/Y chromosome haplogroups, and comparing ancestry estimates between chr1-22 and chrX. This suggested that most of the Persian ancestry was from males and most of the African ancestry was from females. The authors methodology to answering these questions is appropriate and their conclusions are well supported. Their dataset is highly novel and the questions they address give this manuscript a high impact on the fields of population genetics and anthropology. However, prior to publication there are multiple comments that the authors need to address:

1. The qpAdm modeling is very thorough in examining multiple combinations of admixing populations. However, I think that there are multiple areas that should be clarified to help a general audience interpret these results:

a. The supplemental section describing the qpAdm modeling presents many different combinations of populations that were used. However, I found it a little challenging in some cases to determine how some models were rejected or why certain models were broken. It might be helpful to include additional descriptions about the criteria that is used to consider a model broken/rejected or what makes a model a better fit than an alternative working model. For example on lines 600-601, "whereas the Indian population breaks a model with the East Asian populations" Is this due to $P = 0.000000$ in all entries for the column "P value for East Asian population on left, Ulladan on right?"

This is correct. We have edited the text to make this clearer in this case and in related discussions.

b. Lines 216-217: "Any model that does not include all three ancestries can be rejected with high statistical confidence." It is not clear from the main text which table/figure includes this data and what is considered high statistical confidence.

In the supplementary information we describe that we cannot fit a two-way model of distinct ancestries for the ancient groups (meaning they must have more than two sources of ancestry). We perform a series of tests to demonstrate that at least three general ancestry types (African, Persian, and Indian) are required for a fitting model. We show that a number of such 3-way models fit statistically ($p > 0.05$). We have now added to the main paper a pointer to the section of the supplement presenting this argumentation.

c. What is the significance of the three lines that are shaded in grey in Extended Data Table 2? Also the same for table S7.

In Extended Data Table 2, the shaded grey lines represent individuals for whom the qpAdm model does not fit with statistical confidence ($p < 0.01$), and we now note this in the legend. In Table S7, the shaded grey lines represent African groups producing fitting models. We have clarified this as well.

d. What is the "POP" column in Table S7?

We have fixed this column header to read "Coefficient for the Ancient African population".

2. How was $K = 8$ chosen as the ADMIXTURE run to present in the main text? Was cross validation considered or were other criteria applied for selecting this value of K ? What criteria did you use for selecting one of the four replicates to represent each value of K ?

In all cases, we used low cross-validation errors, high log likelihood scores, and a low number of reference populations so as to not overfit for choosing K for the ADMIXTURE analysis. We now describe this in the legends.

3. How was the relatedness filtering performed on the new ancient dna samples? Could you include a short description in the methods section describing how relatedness was determined?

The method is described in Kennet et al. Nature Communications 2017, and is very similar to the READ method (Monroy Kuhn et al. PLoS One 2018). We now reference both.

4. Main text lines 153-154: "We also radiocarbon dated enough individuals from each site to establish chronology" Can you clarify how many samples are meant by "enough?"

We edited this sentence to remove "enough" since the dated individuals are shown in table 1. What we had meant is that we dated 3 individuals from Makwasinyi which we judged was adequate given their high level of genetic homogeneity and the consistency of their dates. We also dated all individuals from Songo Mnara since there was relatively greater genetic variation there (meaning that it was plausible to think that the individuals could have lived at very different times). We then dated 10 individuals from Mtwapa and 5 from Manda until we obtained a relatively comprehensive picture of where the site fell in the chronology of all the sites we investigated in this paper. We had dates from two labs, PSUAMS and ISGS, which differed significantly in their means. At Mtwapa, the ISGS dates are systematically and significantly younger. At Manda, we don't have enough dates of both types to support a statistical test, but the ISGS is the youngest of all six. As a result, we removed all ISGS dates as we deemed them less accurate.

5. Main text lines 330-332: "Under the simplifying assumption that the mixture occurred in a single event over just a few generations, rather than over a period that spanned many generations" How would your results be impacted if this assumption is incorrect?

What we estimate directly is the proportion of X chromosomes and autosomes that come from African and Asian ancestors. Translating this to proportions of ancestry from male and female ancestors requires making assumptions about the dynamics of the mixture, which is affected by variables such as whether the mixture occurred over a relatively short period or was drawn out over time, and whether there was differential treatment of male and female offspring of parents of different ancestry with regard to the degree to which they were incorporated into the population. We were not able to determine the dynamics, so we made a simplifying assumption and were clear about that assumption. In our revision, we clarify what this limitation means, writing: "The comparison of chromosome X and the autosomes allows us to conclude confidently that regardless of the social dynamics of mixture, Persian males and African females had a greater genetic impact on the mixed population than did Persian females and African males, respectively."

6. On line 379 in the main text: "After removing two outliers (according to the ADMIXTURE graph." I think this is described in the supplement, but it is not clear from the main text what an outlier sample is from your ADMIXTURE analysis and how that was determined.

We have marked the outliers in the ADMIXTURE with an "x" and better explained the determination.

7. Main text line 882: "haplogroups were determined according to the Yfull tree" Is there a citation for "Yfull tree?"

We have now included the source json site for Yfull and we have cited a paper that describes the methodology we used for determining Y haplogroups.

8. Bellow are a few grammar/spelling comments to clarify:

a. Supplement line 652: "ancestry, degraded the model generally by one two orders of magnitude." Should this be "one or two?"

We have fixed this.

b. Supplement line 1144: "Kilwa individual to use as a reference, and so we used archaeological context dte range of 1300." I think "dte" should be "date."

We have fixed this.

Referee #3

A. The paper outlines the outcomes of the genetic analysis of 80 individuals from six sites across Kenya and Tanzania - of which 54 were included in the analysis. These individuals are associated with dates 1300-1800CE. These are compared to 13 samples from Makwasinyi, from 1650-1950CE considered as a proxy for contemporary inland communities. The data suggest some differences between the northern Swahili coast and Southern Swahili coast - with those individuals from northern sites showing African/Persian/Indian ancestry; while individuals from southern sites showing Arabic ancestry - particularly those from later periods. Individuals from Kilwa Songo Mnara and Lindi return strong Bantu signature, while those from northern sites the African component appears to be a mixture of Bantu and Pastoralist-related ancestry - consistent with the Makwasinyi samples.

Analysis suggests that foreign DNA was likely introduced through the males, with male ancestors mostly from Persia, and female ancestors mostly African or Indian.

B. The study demonstrates a novel approach to examining the question of Swahili origins and represents a significant new body of work.

C. The study presents a reasonable cumulative dataset, however it is heavily skewed to Mtwapa over the other sites assessed. The scale of the assemblages is also slightly confusing. The paper introduces a sample of 80 individuals. They then remove 26 for various reasons (poor samples, relatives to others etc). It seems 67 individuals provided radiocarbon dates. They note dated samples represent 24 (44%) of the 54 remaining individuals.

In figure 1 51 dated samples are shown 10 from Makwasinyi and the remaining 41 from the other assemblages. In figure 2 the 10 from Makwasinyi were excluded. However, I am not clear what happened to the other 13 samples from individuals of sufficient quality for assessment. I initially thought this was because the individuals presented in the figures had dates associated but there seem to be too many then. This is confusing.

We have added clarifications to the legends. In the legend of Figure 1, we added the following description of our date ranges (from 26 direct radiocarbon dates at the various sites and 1 date range determined from archaeological context):

"Sites with ancient DNA are marked with black shapes and labeled on the zoom-in inset with (X|Y), where X is the number of individuals with data, and Y the number of individuals for which we report high-resolution analyses. Chronology is given as the union of 95% confidence intervals for direct radiocarbon dates on the skeletons rounding to the nearest 50 years and denoted with calCE, and CE for sites with only archaeological context."

We also clarify how many individuals are dated in the main text and direct the reader to Extended Data Table 1 with all the radiocarbon dates listed. We have also added several new PSUAMS radiocarbon dates in our revision, but we have removed some ISGS dates as we found them to deviate significantly

from the PSUAMS dates and became worried about introducing a lab-based artifact. We updated the number of radiocarbon dates reported as newly generated from 24 to 26 in the Data set overview section.

I do think more transparency is needed as to why the majority of the individuals selected post date 1300. Analysis of earlier populations would have allowed the authors to explore whether or how genetic ancestry changed once there was greater connection with the Indian Ocean World.

We would have loved to be able to include earlier individuals in our study, but unfortunately the available earlier ancient DNA data from this region is limited, and we were not successful in ancient DNA analysis of those skeletal individuals from earlier periods we attempted. To more accurately paint a picture of the current state of ancient DNA from the East African coastal region, we have now added in the following paragraph to the Data set section:

“Ancient DNA data from four individuals from the eastern African coast has previously been published, but none is from a Swahili town [32]. An individual from ~1400 CE whose remains were recovered from Makangale Cave on Pemba Island predominantly had ancestry related to western African groups (an ancestry common today in speakers of Bantu languages and prevalent in eastern Africa; henceforth called ‘Bantu-associated’) [32]. Another individual from Makangale Cave on Pemba Island dated to ~600 CE, an individual from ~600 CE from Kuumbi Cave on Zanzibar Island, and an individual from ~1500 CE from Panga ya Saidi in Kenya, all predominantly had sub-Saharan African forager-associated ancestry [32]. There is no indication of Eurasian ancestry deriving from migrations in the last 2000 years in any of these individuals, which differs from the ancestry pattern in the individuals from the medieval coastal towns of this study.”

The methods all seem appropriate, but do assume a lot of prior knowledge of genetic analysis - this may be appropriate for the readership, but it is quite jargon heavy. The data analysis is quite dense which can make it very difficult to read. This is particularly apparent in the section on genetic ancestry by Sex paragraphs lines 284-236. Thinning out and making the methods more accessible to a wider readership seems important as this paper is likely to have appeal not just to those working on Genetics, but those looking at the archaeology the East African coast more broadly.

In our revision we sought to make the main text accessible to a wider disciplinary audience. We added explanatory sentences or even paragraphs throughout and simplified language where possible.

I think clarity may be slightly impacted by the decision to present the ancestral groupings overall rather than the data from each of the sites individually first. I think this would have helped tease out both the chronological and regional variation more clearly. In the current form you jump around from site to site - which can be hard to follow - especially if you didn't already have a clear knowledge of where the sites are.

We have edited the first few paragraphs of the results with the goal of making the findings more accessible for a broader readership. For the revision, we take the approach of beginning by providing a broad overview of the ancestry patterns at all sites, as we think this is most useful for the wide range of readers who are interested in the common patterns across sites and may not be as familiar with the site-by-site archaeological differences.

Line 270/271 - I think you've mixed up the dark yellow and light yellow in your caption?

Thank you, we have now fixed this.

In the data section the authors reference two previous studies Raaum et al [29] and Brucato et al [28]. I think the work could benefit from a summary of these studies as it is not clear how comparable they are.

We agree that the qualitative differences in genetic patterns we reported are interesting. For this revision, we have explored this even more deeply by adding new experiments which generated whole genome data from the individuals analyzed in Raaum et al. (prior to our new data collection there was no whole genome data from the Raaum et al. individuals). The sampling strategies for the two studies are different, with the Raaum et al. individuals ascertained based on their self-identifying as having cultural connection to the Swahili coastal towns for multiple generations going back in time; in contrast, the Brucato et al. individuals were ascertained based on identifying as Swahili language speakers (but not necessarily with a cultural connection to the Swahili coastal towns and civilization). Our genome-wide results confirm the qualitatively different ancestry profiles for the two groups which was suggested by the Y chromosome results we previously reported. The Raaum et al. dataset shows a greater persistence of the medieval ancestry profile (as seen for example at Mtwapa and Manda) than does the Brucato et al. datasets.

D. Statistics and treatment of uncertainties are clear, though I do think the explanation for some of the outliers as indicative of the cosmopolitan nature of the region is a bit over general.

We no longer use the term “cosmopolitan” and are more conservative in our revision in our statements about what can be concluded from the study about connectedness of people.

E. The conclusions draw on a wider range of sources (historical, linguistic, etc) to interpret the datasets presented. The limitations of the study - in terms of the selection of burials from elite contexts and associated with Islamic burials are clear.

F. None additional to the comments above.

G. References are appropriate.

H. The abstract is clear and appropriate, though it does suggest a larger sample than is actually included in the analysis. The introduction and conclusions are clear.

Thank you for the positive review of our work.

Reviewer Reports on the First Revision:

Referees' comments:

Referee #1 (Remarks to the Author):

I reviewed the initial submission of this manuscript, and I am satisfied with the revised version. I was pleased to see that the authors generated and added nuclear data from a greater number of present-day individuals. This dataset represents a major contribution for further studies.

The authors addressed all my concerns and reworked the main text by providing clarifications and adding supplementary notes when required by the reviewers. The lecture of the manuscript is more pleasant, and the figures are clearer and more to the point.

The study is relevant and gives insights about the human population history of an under-studied part of the world. I, personally, find this study very interesting.

The dataset generated will be of interest for further studies on African population history. The statistics used in the study are all appropriate and presented clearly. The authors treated the uncertainties cautiously and made an effort to add clarifications in the main and supplementary texts when requested by the reviewers.

Authors' interpretations are logical, and their conclusions are solid. I do not detect any scientific flaws in the manuscript which in my opinion is acceptable for publication.

Referee #2 (Remarks to the Author):

The authors sufficiently responded to and addressed all of my comments in this revised manuscript. Their main conclusions are well supported by their results, and are clear to understand. I do not have any additional comments for them to address.

Referee #3 (Remarks to the Author):

A. This paper outlines the DNA analysis of 80 individuals from six archaeological sites along the East African coast (Kenya and Tanzania). A further 13 samples were analysed from Makwasinyi which served as a proxy for inland African groups that would have interacted with the coastal communities. Data was compared to that collected by two other studies of present day populations that identified as Swahili. Key findings are: a) DNA analysis is consistent with historical, archaeological and linguistic data which suggests admixture of African and Foreign (primarily Persian-associated) ancestry by c. 1000CE. b) analysis of ancestry related to genetically determined sex suggests the male ancestry was predominantly from Persia, with female ancestry mainly from Africa and India. The data suggests some variations between the Kenyan and Tanzanian samples, with those individuals from northern sites showing African/Persian/Indian ancestry; while individuals from southern sites showing Arabic ancestry -

particularly those from later periods. Individuals from Kilwa Songo Mnara and Lindi return strong Bantu signature, while those from northern sites the African component appears to be a mixture of

Bantu and Pastoralist-related ancestry - consistent with the Makwasiyi samples.

B. The study demonstrates a novel approach to examining the question of Swahili origins and represents a significant new body of work.

C. The data presented in the papers has benefitted from revision with greater clarity in the scale and limitations of the data more explicitly defined. The methods are appropriate and have been described more generally making them easier for non-specialists to follow. The presentation of the data is structured to document the evidence used to support the different interpretations posed - drawing of different forms of analysis. This is in general quite clear, but the authors could go further in being explicit about the scale of the samples they are referring to in each context - as this varies according to the approach they adopt or the question they seek to answer.

D. The statistics appear appropriate for the work presented. While the authors are more circumspect in some of their interpretations in the current submission, I think they could go further in discussing the outliers: how many are there and why do they not fit the pattern? This section is very short compared with the rest of the data and is still explained away by a very general statement. The limitations outlined in the discussion I think could be more usefully included earlier in the paper as its current placement immediately before the concluding paragraph seems to put a downer on the rest of the results discussed.

E. The discussion provides context for the genetic data in relation to other strands of evidence. It would also be helpful to link some of the data back into the wider context of the Swahili outlined in the much improved introductory sections., however I do feel like the paper could benefit from a stronger conclusion. As it is it seems to peter out.

F. None additional to those outlined above.

G. References are appropriate

H. The abstract is appropriate. Introduction is much clearer. See note above regarding the conclusions.

Author Rebuttals to First Revision:

Referee #3 point C: “the authors could go further in being explicit about the scale of the samples they are referring to in each context - as this varies according to the approach they adopt or the question they seek to answer”

In our revision, we have added multiple clauses point out when small sample sizes affect our ability to make precise inferences. Here are several sentences highlighting this:

“Some coastal individuals, particularly from Songo Mnara and Lindi, do not fall on this cline, suggesting additional complexity, although our power to understand this variability is limited by small sample sizes.”

“We could not perform the same analysis at Songo Mnara because no individuals with high quality data fit the three-way model.”

“Our finding of coastal individuals who differ compared to the others from similar times or regions attest to continued exchange with people in the Indian Ocean trading network, although our sample size is too small to identify general patterns.”

Referee #3 point D: “I think they could go further in discussing the outliers: how many are there and why do they not fit the pattern? This section is very short compared with the rest of the data and is still explained away by a very general statement.

We have moved the previous section on outliers to an earlier section of the manuscript where it fits better.

The limitations outlined in the discussion I think could be more usefully included earlier in the paper as its current placement immediately before the concluding paragraph seems to put a downer on the rest of the results discussed.”

We have integrated some notes on the limitations earlier in the manuscript. We also shortened the penultimate paragraph, which now focuses on biases in the sampling strategy and provides guidance on how to interpret but not overinterpret the results in light of them.

E. The discussion provides contextualises the genetic data in relation to other strands of evidence. It would also be helpful to link some of the data back into the wider context of the Swahili outlined in the much improved introductory sections., however I do feel like the paper could benefit from a stronger conclusion. As it is it seems to peter out.

We now have a stronger final paragraph:

“These findings highlight multiple directions for future ancient DNA work. One is to study individuals pre-dating the 12th century, including before and after the major populations mixtures we show occurred around 1000 CE. Another is to study individuals from unsampled parts of the Swahili world, including the present-day countries of Somalia, Mozambique, the Comoros Islands, and Madagascar. The results presented here, however, provide unambiguous evidence of ongoing cultural mixing on the East African coast for more than a millennium, in which African people interacted and had families with immigrants from other parts of Africa and the Indian Ocean world. Narratives of ancestry have a

complex history on the eastern African coast, and the genetic findings of long-standing, sex-biased mixtures add to this complexity.”